# Towards Open-World Feature Extrapolation: An Inductive Graph Learning Approach

**Qitian Wu, Chenxiao Yang, Junchi Yan**[*]
Department of Computer Science and Engineering
MoE Key Lab of Artificial Intelligence, AI Institute
Shanghai Jiao Tong University
{echo740, chr26195, yanjunchi}@sjtu.edu.cn

## Abstract

We target open-world feature extrapolation problem where the feature space of input data goes through expansion and a model trained on partially observed features needs to handle new features in test data without further retraining. The problem is of much significance for dealing with features incrementally collected from different fields. To this end, we propose a new learning paradigm with graph representation and learning. Our framework contains two modules: 1) a backbone network (e.g., feedforward neural nets) as a lower model takes features as input and outputs predicted labels; 2) a graph neural network as an upper model learns to extrapolate embeddings for new features via message passing over a feature-data graph built from observed data. Based on our framework, we design two training strategies, a self-supervised approach and an inductive learning approach, to endow the model with extrapolation ability and alleviate feature-level over-fitting. We also provide theoretical analysis on the generalization error on test data with new features, which dissects the impact of training features and algorithms on generalization performance. Our experiments over several classification datasets and large-scale advertisement click prediction datasets demonstrate that our model can produce effective embeddings for unseen features and significantly outperforms baseline methods that adopt KNN and local aggregation.

## 1 Introduction

Learning a mapping from observation $\mathbf{x}$ (a vector of attribute features) to label $y$ is a fundamental and pervasive problem in ML community, with extensive applications spanning from classification/regression tasks for tabular data to advertisement click prediction [23, 38, 13, 37, 27], item recommendation [26, 52, 14, 53], question answering [12, 11], or AI more broadly. Existing approaches focus on a fixed input feature space shared by training and test data. Nevertheless, practical ML systems interact with a dynamic open-world where features are incrementally collected. For example, in recommender/advertisement systems, there often occur new user profile features unseen before for current prediction tasks. Also, with the advances of multi-modal recognition [45] and federated learning [29], it is a requirement for a model trained with partial features to incorporate new features from other fields for decision-making on a target task.

A challenge stems from the fact that off-the-shelf neural network models cannot deal with new features without re-training on new data. As shown in Fig. 1(a), a neural network builds a mapping from input features to a hidden representation through a weight matrix in the first layer. Given new features as input, the network needs to be augmented with new weight parameters and cannot map the new features to a desirable position in latent space if without re-training.

---

[*]Junchi Yan is the corresponding author.

35th Conference on Neural Information Processing Systems (NeurIPS 2021).

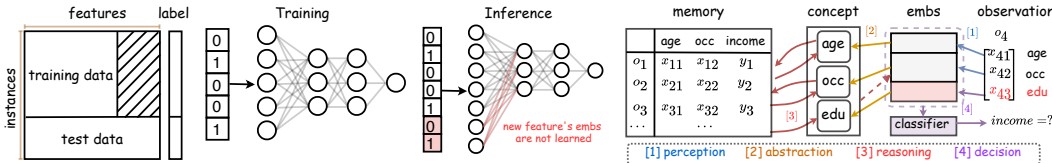

(a) Limitation of neural network models for open-world feature extrapolation
(b) A new approach inspired by human's thinking

Figure 1: (a) The open-world feature extrapolation problem is defined over training data with partial features and test data with augmented feature space. Existing neural models cannot deal with new features without re-training. (b) Oue approach has an analogy to thinking process in human's brain. Imagine a person trained on using features *age*, *occ* to predict one's *income* and tested on test cases with a new feature *edu*. There are four processes from acquiring observation to giving final prediction.

However, model re-training would be tricky and bring up several issues. First, re-training a model with both previous and new features would be highly time-consuming and cannot meet the requirement for online systems. Alternatively, one can re-train a trained model only on new features, which will induce risks for over-fitting new data or forgetting previous data.

Different from machines, fortunately, humans are often equipped with the ability for extrapolating to unseen features and distill the knowledge in new information for solving a target task without any re-training on new data. The inherent gap between existing ML approaches and human intelligence raises a research question: *Can we design a ML model that is trained on one set of features and able to generalize to combine new unseen features for the same task without further training?*

To find desirable solutions to this non-trivial question is challenging. Let us resort to how humans think and act in physical world scenarios. Imagine that we are well trained on predicting a person's income with one's age and occupation from historical records. Now we are asked to estimate the income of a new person with his/her age, occ. and education (a new feature). We can modularize the process in brain systems that distill and leverage the knowledge in new observations into four steps, as shown in Fig. 1(b). 1) *Perception*: the observations are recognized by our senses; 2) *Abstraction:* the perceived information is aligned with concepts in our cognition; 3) *Reasoning:* we search similar concepts and observations in our memory to understand and absorb the new knowledge; 4) *Decision:* with new understanding and abstraction, we make final decisions for the prediction.

The above human's thinking process inspires the methodology in our paper where we propose a new learning paradigm for open-world feature extrapolation. Our proposed framework contains two modules: a backbone network, which can be a feedforward neural network, and a graph neural network. 1) The backbone network first maps input features to embeddings, which can be seen as *perception* from observation. 2) We then treat observed data matrix (each row represents a feature vector for one instance) as a feature-data bipartite graph, which explicitly define proximity and locality structures among features and instances. Then a graph neural network is harnessed for neural message passing over adjacent features and instances in a latent space (*abstraction*). 3) The GNN will inductively compute embeddings for new features based on those of existing ones, mimicking the *reasoning* process from familiar concepts to new ones in our brain. 4) The newly obtained embeddings that capture both semantics and feature-level relations will be used to obtain hidden representations of new data with unseen features and make final *decisions*.

To endow the model with ability for extrapolation to new features, we propose two training algorithms, one by self-supervised learning and one by inductive learning. The proposed model and its learning algorithm can easily scale to large-scale dataset (with millions of features and instances) via mini-batch training. We also provide a theoretical analysis on the generalization error on test data with new features, which allows us to dissect the impact of training features and algorithms on generalization performance. To verify our approach, we conduct extensive experiments on several real-world datasets, including six classification datasets from biology, engineering and social domains with diverse features, as well as two large-scale datasets for advertisement click prediction. Our model is trained on training data using partial features and tested on test data with a mixture of seen and unseen features. The results demonstrate that 1) our approach consistently outperform models not using new features for inference; 2) our approach achieves averagely $29.8\%$ higher Accuracy than baseline methods using KNN, pooling or local aggregation for feature extrapolation; 3) our approach even exceeds models using incremental training on new features, yielding average $4.1\%$ higher Accuracy.

**Our contributions are:** 1) We formulate open-world feature extrapolation problem and show that it is feasible to extend neural models for extrapolating to new features without re-training; 2) We propose a new graph-learning model and two training algorithms for feature extrapolation problem; 3) Our theoretical analysis shows that the generalization error for data with new features relies on the number of training features and the randomness in training algorithms; 4) We conduct comprehensive experiments and show the effectiveness, applicality and scalability of proposed method.

## 2 Methodology

We focus on attribute features as input in this paper. An input instance is a vector $\mathbf{r}_i = [r_{im}]_{m=1}^d \in \mathbb{R}^d$ where each entry $r_{im}$ denotes a *raw feature* (like age, occ., edu., etc.). If $r_{im}$ is a discrete/categorical raw feature with $R_m$ possible values, its space is an integer set $\{0, 1, \cdots, R_m - 1\}$. If $r_{im'}$ is a continuous one, a common practice is to convert it into a discrete feature within space $\{0, 1, \cdots, R_{m'} - 1\}$ by evenly division [24] or log transformation [23]. We call $R_m$ as *cardinality* for $m$-th raw feature.

An effective way to handle attribute features is via one-hot encoding [13, 24, 17]. For $r_{im}$ with cardinality $R_m$ we convert it into a $R_m$-dimensional one-hot vector $\mathbf{x}_i^m$ where the unique 1 indexes the value. In this way, one can convert an input $\mathbf{r}_i$ into a concatenation of one-hot vectors:

$$\mathbf{x}_i = \left[\mathbf{x}_i^1, \mathbf{x}_i^2, \cdots, \mathbf{x}_i^d\right], \text{ where } 1 \leq \forall m \leq d, \ \mathbf{x}_i^m \in \{0, 1\}^{R_m} \text{ is a one-hot vector.} \tag{1}$$

Use $x_{ij}$ to denote the $j$-th entry of $\mathbf{x}_i$ and we call each $x_{ij}$ as *feature* in this paper. Assume $D = \sum_{m=1}^d R_m$ denotes the number of features and, as a reminder, $d$ is the number of raw features.

We next give a formal definition for open-world feature extrapolation problem in this paper: given training data $\{(\mathbf{x}_i, y_i)\}_{i \in I_{tr}}$ where $\mathbf{x}_i \in \mathcal{X}_{tr} = \{0, 1\}^D$, $y_i \in \mathcal{Y}$ and $I_{tr}$ is a set of indices, we aim to learn a model that can generalize to test data $\{\overline{\mathbf{x}}_{i'}, y_{i'}\}_{i' \in I_{te}}$ where $\overline{\mathbf{x}}_{i'} \in \mathcal{X}_{te} = \{0, 1\}^{\overline{D}}$, $y_{i'} \in \mathcal{Y}$ and $I_{te}$ is another set of indices. We term $\mathcal{X}_{tr}$ as *training feature space* and $\mathcal{X}_{te}$ as *test feature space*. We assume 1) the label space $\mathcal{Y}$ is shared by training and test data, and 2) $\mathcal{X}_{tr} \subset \mathcal{X}_{te}$, i.e., test feature space is an extension of training feature space. The feature space expansion stems from two possible causes: 1) there appear new raw features incrementally collected from other fields (i.e., $d$ increases) or 2) there appear new values out of the known space of existing raw features (i.e., $R_m$ increases).

### 2.1 Proposed Model

Our model contains three parts: 1) feature representation that builds a bipartite feature-data graph from input data; 2) a backbone network which is essentially a neural network model that predicts the labels when fed with input data; 3) a GNN model that inductively compute features' embeddings based on their proximity and local structures to achieve feature extrapolation.

**Feature Representation with Graphs.** We stack the feature vectors of all the training data as a matrix $\mathbf{X}_{tr} = [\mathbf{x}_i]_{i \in I_{tr}} \in \{0, 1\}^{N \times D}$ where $N = |I_{tr}|$. Then we treat each feature and instance as nodes and construct a bipartite graph between them. Formally, we define a node set $F_{tr} \cup I_{tr}$ where $F_{tr} = \{f_j\}_{j=1}^D$ with $f_j$ the $j$-th feature and $I_{tr} = \{o_i\}_{i=1}^N$ with $o_i$ the $i$-th instance in training set. The binary matrix $\mathbf{X}_{tr}$ constitutes an adjacency matrix where the non-zero entries indicate edges connecting two nodes in $F_{tr}$ and $I_{tr}$, respectively. The induced feature-data bipartite graph will play an important role in our extrapolation approach. The representation is flexible for variable-size feature set, enabling our model to handle test data $\overline{\mathbf{x}}_{i'} \in \{0, 1\}^{\overline{D}}$ which gives $\mathbf{X}_{te} = [\overline{\mathbf{x}}_{i'}]_{i' \in I_{te}}$.

**Backbone Networks.** We next consider a prediction model $h_\theta(\cdot)$ as a backbone network that maps data features $\mathbf{x}_i$ to predicted label $\hat{y}_i$. Without loss of generality, a default choice for $h_\theta$ is a feedforward neural network. The first layer serves as an embedding layer which shrinks $\mathbf{x}_i$ into a $H$-dimensional hidden vector $\mathbf{z}_i = \mathbf{x}_i \mathbf{W}$ where $\mathbf{W} \in \mathbb{R}^{D \times H}$ denotes a weight matrix. The subsequent network (called *classifier*) is often a stack of neural layers that predicts label $\hat{y}_i = \text{FFN}(\mathbf{z}_i; \phi)$. We use the notation $\hat{y}_i = h(\mathbf{x}_i; \phi, \mathbf{W})$ to highlight two sets of parameters and $\theta = [\phi, \mathbf{W}]$.

Notice that the matrix multiplication in the embedding layer is equivalent to a two-step procedure: 1) a lookup of feature embeddings and 2) a permutation-invariant aggregation. More specifically, we consider $\mathbf{W}$ as a stack of weight vectors $\mathbf{W} = [\mathbf{w}_j]_{j=1}^D$ where $\mathbf{w}_j \in \mathbb{R}^{1 \times H}$ corresponds to the embedding of feature $f_j$. The non-zero entries in $\mathbf{x}_i$ will index the corresponding rows of $\mathbf{W}$ and induce a set of embeddings $\{\mathbf{z}_i^m\}_{m=1}^d$ where $\mathbf{z}_i^m$ is the embedding given by $\mathbf{x}_i^m$ (i.e., one-hot vector

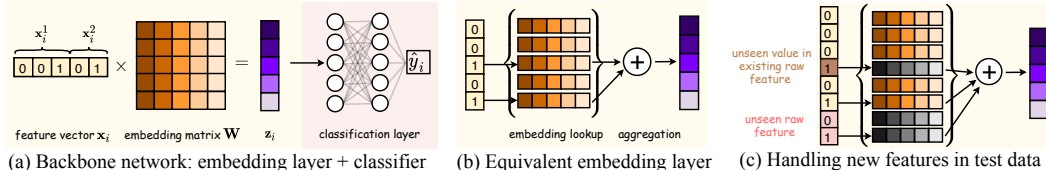

(a) Backbone network: embedding layer + classifier  (b) Equivalent embedding layer  (c) Handling new features in test data

Figure 2: An illustration for the feasibility of extrapolation for new feature space with neural network models. (a) The backbone network is a feedforward neural network that is fed with input feature vector $\mathbf{x}_i$ and outputs predicted label $\hat{y}_i$. The first layer can be seen as an embedding layer where an embedding matrix is multiplied with the input vector to obtain an intermediate hidden representation $\mathbf{z}_i$. (b) The embedding layer can be equivalently replaced by an embedding lookup and an aggregation over indexed feature embeddings. The aggregation is permutation-invariant w.r.t. the order of input features. (c) The permutation-invariant property enables the model to handle variable-length input feature vector (new features may come from unseen values of existing raw features or unseen raw features). The problem boils down to learning new features' embeddings.

of $m$-th raw feature). Then the hidden vector of $i$-th instance can be obtained by aggregation, i.e., $\mathbf{z}_i = \sum_{m=1}^{d} \mathbf{z}_i^m$ which is permutation-invariant w.r.t. the order of feature embeddings in $\{\mathbf{z}_i^m\}_{m=1}^{d}$. A more intuitive illustration is presented in Fig. 2.

The permutation-invariant property opens a way for handling variable-length feature vectors $\overline{\mathbf{x}}_{i'}$ [56]. Essentially, on condition that we have embeddings for input features, we can add them up to get a fixed-dimensional hidden representation $\mathbf{z}_{i'}$ as input for the subsequent classifier. Therefore, the problem boils down to learning feature embeddings, especially *how to extrapolate for embeddings of new features based on those of existing ones*.

*Remark.* Instead of using sum aggregation, some existing architectures consider concatenation of $d$ embedding vectors $\mathbf{z}_i^m$'s, which is essentially equivalent to sum aggregation (see Appendix A for more details). Therefore, the permutation-invariant property holds for widely adopted deep models.

**GNN for Feature Extrapolation.** We proceed to propose a graph neural networks (GNN) model for embedding learning with the feature-data graph. Our key insight is that the bipartite graph explicitly embodies features' co-occurrence in observed instances, which reflects the proximity among features. Once we conduct message passing for feature embeddings over the graph structures, the embeddings of similar features can be leveraged to compute and update each feature's embedding. The model can learn to extrapolate for new features' embeddings using those of existing features with locality structures in a data-driven manner. The message passing over the defined graph representation is *inductive* w.r.t. variable-sized feature nodes and instance nodes, which enables the model to tackle new feature space with distinct feature sizes and supports.

Specifically, we consider the embeddings $\mathbf{w}_j$ as an initial state $\mathbf{w}_j^{(0)}$ for node $f_j$ in $\mathcal{F}_{tr}$. The initial states of instance nodes are set as zero vectors with equal dimension as the feature nodes, i.e. $\mathbf{s}_i^{(0)} = \mathbf{0}$. The interaction between two sets of nodes $\{\mathbf{w}_j\}_{j=1}^{D}$ and $\{\mathbf{s}_i\}_{i=1}^{N}$ can be modeled via graph neural networks where the node states in the $l$-th layer are updated by

$$\mathbf{s}_i^{(l)} = \mathbf{P}^{(l)}\text{COMB}\left(\mathbf{s}_i^{(l-1)}, \text{AGG}(\{\mathbf{w}_k^{(l-1)}\}|\forall k, x_{ik} = 1)\right),$$
$$\mathbf{w}_j^{(l)} = \mathbf{P}^{(l)}\text{COMB}\left(\mathbf{w}_j^{(l-1)}, \text{AGG}(\{\mathbf{s}_k^{(l-1)}\}|\forall k, x_{jk} = 1)\right),$$

(2)

where $\mathbf{P}^{(l)} \in \mathbb{R}^{H \times H}$ is a weight matrix and we do not use non-linearity since it would degrade the performance empirically. For any new feature $f_{j'}$ in test data, we can set its initial state as a zero vector $\mathbf{w}_{j'}^{(0)} = \mathbf{0}$. The GNN model outputs updated embeddings for feature nodes and we further use them as the feature embeddings in the backbone network. Fig. 3 presents the feedforward computation of proposed model. Formally, with $L$-layer GNN, the GNN network $g$ gives updated feature embeddings $\hat{\mathbf{W}} = [\mathbf{w}_j^{(L)}]_{j=1}^{D} = g(\mathbf{W}, \mathbf{X}; \omega)$ and then the backbone network outputs prediction $\hat{y}_i = h(\mathbf{x}_i; \phi, \hat{\mathbf{W}})$ where $\mathbf{X} = \mathbf{X}_{tr}$ for training, $\mathbf{X} = \mathbf{X}_{te}$ for test and $\omega = \{\mathbf{P}^{(l)}\}_{l=1}^{L}$ denotes $g$'s parameters,.

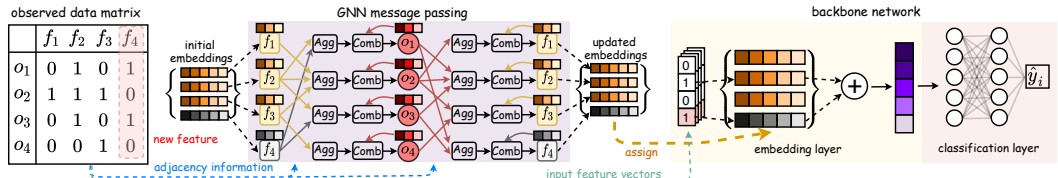

Figure 3: Feedforward of our model with input $\{\mathbf{x}_i\}$: we build a feature-data graph between feature nodes $\{f_j\}$ and instance nodes $\{o_i\}$. A GNN is used to inductively compute features' embeddings via message passing and a backbone network uses the updated embeddings to predict labels $\{y_i\}$.

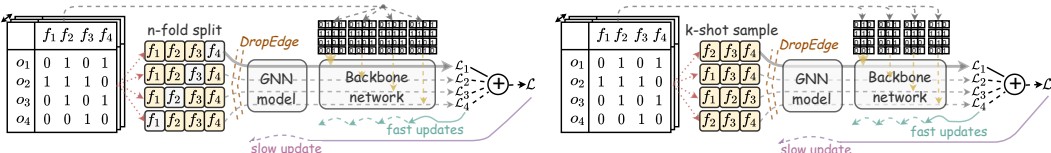

(a) self-supervised learning for feature extrapolation      (b) inductive learning for feature extrapolation

Figure 4: Illustration of two proposed training approaches: (a) self-supervised learning with n-fold splitting ($n = 4$), and (b) inductive learning with k-shot sampling ($k = 3$). The two methods differ in configuration of proxy data, and both consider asynchronous updating rule: the backbone network is updated with $n$-fold proxy data before the GNN network is updated once with an accumulated loss.

## 2.2 Model Learning

We next discuss approaches for model training. In order to enable the model to extrapolate for new features, we put forward two useful strategies. 1) *Proxy training data:* we only use partial features from training set as observed ones for each update. 2) *Asynchronous updates:* we decouple the training of backbone network and GNN network and using different updating speeds for them in a nested manner (see Fig. 4). Based on these, we proceed to propose two specific training approaches.

**Self-supervised Learning with N-fold Splits.** To mimic new features in the future, we can mask some observed features and let the model use the remaining features to estimate the embeddings of the masked ones. For a given feature set $F_{tr} = \{f_j\}_{j=1}^D$ of data $\{\mathbf{x}_i\}_{i \in I_{tr}}$, we consider an n-fold splitting method: in each iteration the features are first randomly shuffled and evenly divided into $n$ disjoint subsets, denoted by $\{\overline{F}_s\}_{s=1}^n$. We then consider asynchronous updating rule for two networks: each iteration contains $n$ times updates for backbone network and one update for GNN model. For the $s$-th update of the backbone, we mask features in $\overline{F}_s$ and set the initial states of masked features as zero vectors before fed into GNN. The GNN network will use the adjacency matrix $\mathbf{X}_{tr}$ to compute updated embeddings for the masked features. The embedding layer will be composed of updated embeddings of masked features and initial embeddings of the remaining features, based on which the backbone network outputs prediction $\hat{y}_i$ for each $\mathbf{x}_i$ and compute the loss function $\mathcal{L}_s = \frac{1}{N} \sum_{i \in I_{tr}} l(y_i, \hat{y}_i)$[2] (where $l(\cdot, \cdot)$ can be cross-entropy for classification). After $n$-step updates for backbone network, we use the accumulated loss $\mathcal{L} = \sum_s^n \mathcal{L}_s$ to update GNN model. The training procedure will repeat the above process until a given time budget.

**Inductive Learning with K-shot Samples.** Alternatively, we can sample over the feature set and only expose partial features to the model for each update. For the $s$-th update, we randomly sample $k$ *raw features*[3] from input data which induces a new feature set $F_s \subset F_{tr}$ and extract the corresponding columns of $\mathbf{X}_{tr} = [\mathbf{x}_i]_{i \in I_{tr}}$ to form a proxy data matrix $\mathbf{X}_s \in \{0, 1\}^{N \times |F_s|}$ (where each instance contains $|F_s|$ features). Then $\mathbf{X}_s$ is fed into GNN to obtain updated embeddings of features in $F_s$, based on which the backbone network outputs prediction for each instance using features in $F_s$.

By contrast, the n-fold splitting contributes to better training stability since the model is updated on each feature in each iteration, while the inductive learning adds more randomness which can help to enhance model's generalization. We will further compare them in our experiments.

**DropEdge for Regularization.** In order to further alleviate over-fitting on training features, we use the DropEdge [40] to regularize our model. We consider a threshold $\rho$ and randomly set nonzero

---

[2]Note that we still use supervised labels for loss though we call this approach as self-supervised learning.

[3]One can also directly sample a certain ratio of features from $F_{tr}$, which might lead to large variance.

entries in $\mathbf{X}_{tr}$ (for self-supervised) or $\mathbf{X}_s$ (for inductive) as zero for each feedforward computation,

$$\tilde{\mathbf{X}}_{tr} = \text{DROPEDGE}(\mathbf{X}_{tr}, \rho) = \{x_{ij} | x_{ij} \in \mathbf{X}_{tr}, e_{ij} > \rho\}, \text{ where } e_{ij} \sim \mathcal{U}(0,1). \tag{3}$$

**Scaling to Large Systems.** To handle prohibitively large datasets for practical systems, we can divide data matrix into mini-batches along the instance dimension. Then, we feed each mini-batch into the model for once model training (including feature-level sampling/splitting, as shown in Fig. 4) or inference. Since the number of nonzero features for each instance is no more than $d$ (a relatively small value), the edge number in each mini-batch will be controlled within $O(Bd)$ (assume a mini-batch contains $B$ instances). Hence, the space cost can be effectively controlled using instance-level mini-batch partition. Yet, note that $B$ could not be arbitrarily small in order to guarantee sufficient message passing over diverse instances. We present the complete training algorithm in Appendix B where the model is trained end-to-end using self-supervised or inductive learning approaches.

## 3    Generalization Analysis

In this section, we analyze the generalization error on test data with new features. We simplify the settings for analysis: 1) the backbone network is a two-layer FNN (an embedding layer $\mathbf{W} \in \mathbb{R}^{D \times H}$ plus a fully-connected layer $\Phi \in \mathbb{R}^{H \times 1}$) with sigmoid output; 2) the GNN network is a $L$-layer GCN which takes mean pooling aggregation over neighbored nodes without linear transformation and non-linearity in each layer; 3) the training algorithm is SGD. With above settings, the model can be written as $\hat{y}_i = \sigma(\sum_{i' \in \mathcal{N}_{\tilde{L}}(i) \cup \{i\}} c_{ii'}^L \mathbf{x}_{i'} \mathbf{W} \Phi)$ where $\mathcal{N}_{\tilde{L}}(i)$ contains all the $\mathbf{x}_{i'}$'s that appear in the $\tilde{L}$-hop neighbors of $\mathbf{x}_i$ in the feature-data graph, $\tilde{L} = 2 \cdot \lfloor \frac{L}{2} \rfloor$, and $c_{ii'}^L \in \mathbb{R}^+$ is a weight that quantifies influence of $\mathbf{x}_i$ on $\mathbf{x}_{i'}$ through $L$-layer mean-pooling graph convolution. More details for the derivation are in Appendix C. Also, we focus our analysis on the case of inductive learning and the results can be extended to self-supervised approach, which we leave for future work.

The data generation process can be described as follows. First, features $f_j$'s are sampled from an unknown distribution $\mathcal{F}$ and form a feature set $F_{tr} = \{f_j\}_{j=1}^D$. Then data $\{(\mathbf{x}_i, y_i)\}_{i \in I_{tr}}$ are sampled from a distribution $\mathcal{D}_{F_{tr}}$ whose support is over $\mathcal{X}_{F_{tr}} \times \mathcal{Y}$, and define $\mathbf{X}_{tr} = \{\mathbf{x}_i\}_{i \in I_{tr}}$ and $Y_{tr} = \{y_i\}_{i \in I_{tr}}$. Using $\psi = [\theta, \omega]$, the model can be denoted by $\hat{Y} = h(\mathbf{X}; \psi)$ (with a simplification from $\hat{Y} = \{\hat{y}_i\}_{i \in I} = \{h(\mathbf{x}_i; \psi)\}_{i \in I}$) with loss function $\mathcal{L}(Y, \hat{Y}) = \frac{1}{|I|} \sum_{i \in I} l(y_i, \hat{y}_i)$.

Recall that in training stage, we randomly partition the instances in $\mathbf{X}_{tr}$ into mini-batches with size $B$ and each mini-batch further samples $k$ raw features to form a feature subset $F_s$. In each update, the model is exposed to a $B \times |F_s|$ sub-matrix from $\mathbf{X}_{tr}$ as proxy training data and uses it for once feedforward and backward computation. With given training data $(\mathbf{X}_{tr}, Y_{tr})$, we define $\mathcal{S}$ as a set of all the proxy data sub-matrices that could be exposed to the model during the training

$$\mathcal{S} = \{(\mathbf{X}_1, Y_1), (\mathbf{X}_2, Y_2), \cdots, (\mathbf{X}_m, Y_m), \cdots, (\mathbf{X}_M, Y_M)\}, \text{ where } M \propto \mathcal{O}\left(\frac{d!}{(d-k)!k!}\right).$$

The training process can be seen as a sequence of operations each of which samples a sub-matrix from $\mathcal{S}$ as proxy data in an i.i.d. manner and computes gradients for one SGD update (more discussions are in Appendix C). Define $A_{\mathcal{S}}$ as a learning algorithm trained on $\mathcal{S}$, which gives a trained model $h(\mathbf{X}; \psi_{\mathcal{S}})$ simplified as $h_{\mathcal{S}}$. The generalization error $R(h_{\mathcal{S}})$ can be defined as

$$R(h_{\mathcal{S}}) = \mathbb{E}_{(\mathbf{X}, Y)} \left[ \mathcal{L}(Y, h(\mathbf{X}; \psi_{\mathcal{S}})) \right], \tag{4}$$

where the expectation contains two stages of sampling: 1) a feature set $F = \{f_j\}$ is sampled according to $f_j \sim \mathcal{F}$, and 2) data $(\mathbf{X}, Y)$ is sampled according to $(\mathbf{x}_i, y_i) \sim \mathcal{D}_F$. The empirical risk that our approach optimizes with the training data would be

$$R_{emp}(h_{\mathcal{S}}) = \frac{1}{M} \sum_{m=1}^M \mathcal{L}(Y_m, h(\mathbf{X}_m; \psi_{\mathcal{S}})). \tag{5}$$

We study the expected generalization gap

$$\mathbb{E}_A[R(h_{\mathcal{S}}) - R_{emp}(h_{\mathcal{S}})], \tag{6}$$

where the expectation is taken over the randomness of $A_{\mathcal{S}}$ stemming from sampling for SGD updates.

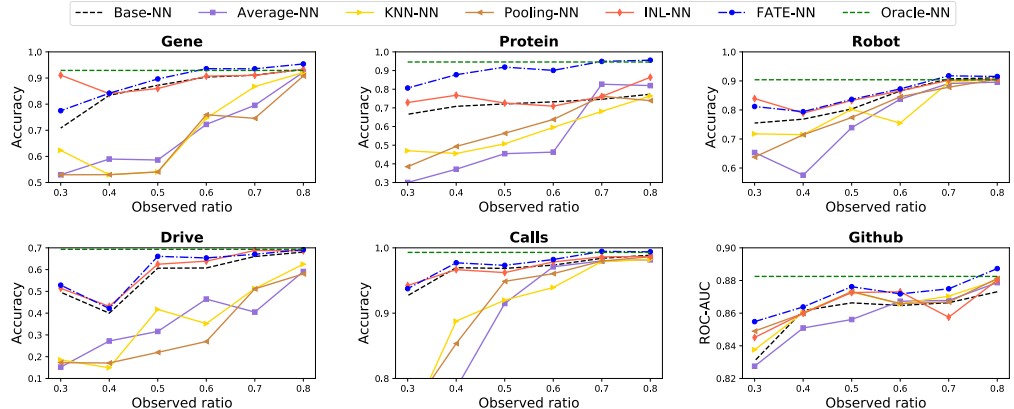

Figure 5: Accuracy/ROC-AUC results for UCI datasets with $30\% \sim 80\%$ observed features for training. We run each experiment five times with different random seeds and report averaged scores.

We assume that the loss function $l(y_i, \hat{y}_i)$ is Lipschitz-continuous and smooth w.r.t. the model output $\hat{y}_i$. Concretely, we have 1) $|l(y, f(\cdot)) - l(y, f'(\cdot))| \leq \beta |f(\cdot) - f'(\cdot)|$ and 2) $|\nabla l(y, f(\cdot)) - \nabla l(y, f'(\cdot))| \leq \beta' |\nabla f(\cdot) - \nabla f'(\cdot)|$. Such condition can be satisfied by widely used loss functions such as cross-entropy and MSE. Then we have the following result (see Appendix C for proof).

**Theorem 1.** *Assume the loss function is bounded by $l(y_i, \hat{y}_i) \leq \lambda$. For a learning algorithm trained on data $\{\mathbf{X}_{tr}, Y_{tr}\}$ with $T$ iterations of SGD updates, with probability at least $1 - \delta$, we have*

$$\mathbb{E}_A[R(h_\mathcal{S}) - R_{emp}(h_\mathcal{S})] \leq \mathcal{O}(\frac{d^T}{M}) + \left(\mathcal{O}(\frac{d^T}{M^2}) + \lambda\right)\sqrt{\frac{\log(1/\delta)}{2M}}. \quad (7)$$

The generalization gap depends on the number of raw features in training data and the size of $\mathcal{S}$. The latter is determined by configuration of proxy training data, particular, sampling over training features. If the sampling introduces more randomness (e.g. $k \approx d/2$), $\mathcal{S}$ would become larger, contributing to tighter gap. However, a large $\mathcal{S}$ would also lead to large variance in training and amplify optimization error. Therefore, there exists a trade-off w.r.t. how to sample/split observed features in training stage. Furthermore, the generalization gap also depends on $d$, and a larger $d$ would result in looser bound (since one often has $T > d$). This is because a larger $d$ would require to deal with more features. As the network becomes *wider* and complex, it would be more prone for over-fitting.

## 4   Experiments

We apply our model FATE (for FeATure Extrapolation Networks) on real-world datasets. First, we consider six classification datasets from UCI Machine Learning Repository [1]: Gene, Protein, Robot, Drive, Calls and Github, as collected from domains like biology, engineering and social networks. The feature numbers are ranged from 219 to 4006 and the instance numbers vary from 1080 to 58509. We consider two large-scale datasets Avazu and Criteo from real-world online advertisement system whose goal is to predict the Click-Through Rate (CTR) of exposed advertisement to users. The two datasets have $\sim 40$ million clicking/non-clicking records as instances and $\sim 2$ million features. More dataset information and implementation details are in Appendix D and E, respectively. The implementation codes are public available at `https://github.com/qitianwu/FATE`.

### 4.1   Experiment on UCI Datasets

**Setup.** We randomly split all the instances into training/validation/test data with the ratio 6:2:2. Then we randomly select a certain ratio ($30\% \sim 80\%$) of features as *observed* ones and use the remaining as *unobserved* ones. The model is trained with the *observed* features of training instances and tested with *all* the features of testing data. We adopt Accuracy as metric for datasets with more than two classes (Gene, Protein, Robot, Drive and Calls) and ROC-AUC for Github with two classes.

**Implementation.** We specify FATE in the following ways. 1) Backbone: a 3-layer feedforward NN. 2) GNN: a 4-layer GCN. 3) Training: self-supervised learning with n-fold splits. Several baselines are considered for comparison and their architectures are all specified as a 3-layer feedforward NN. First,

Table 1: ROC-AUC results for 8 test splits (T1-T8) on Avazu and Criteo datasets.

| Dataset | Backbone | Model | T1 | T2 | T3 | T4 | T5 | T6 | T7 | T8 | Overall |
|---|---|---|---|---|---|---|---|---|---|---|---|
| Avazu | NN | Base | 0.666 | 0.680 | 0.691 | 0.694 | 0.699 | 0.703 | 0.705 | 0.705 | $0.693 \pm 0.012$ |
| | | Pooling | 0.655 | 0.671 | 0.683 | 0.683 | 0.689 | 0.694 | 0.697 | 0.697 | $0.684 \pm 0.011$ |
| | | **FATE** | **0.689** | **0.699** | **0.708** | **0.710** | **0.715** | **0.720** | **0.721** | **0.721** | $\mathbf{0.710} \pm 0.010$ |
| | DeepFM | Base | 0.675 | 0.684 | 0.694 | 0.697 | 0.699 | 0.706 | 0.708 | 0.706 | $0.697 \pm 0.009$ |
| | | Pooling | 0.666 | 0.676 | 0.685 | 0.685 | 0.688 | 0.693 | 0.694 | 0.694 | $0.685 \pm 0.009$ |
| | | **FATE** | **0.692** | **0.702** | **0.711** | **0.714** | **0.718** | **0.722** | **0.724** | **0.724** | $\mathbf{0.713} \pm 0.010$ |
| Criteo | NN | Base | 0.761 | 0.761 | 0.763 | 0.763 | 0.765 | 0.766 | 0.766 | 0.766 | $0.764 \pm 0.002$ |
| | | Pooling | 0.761 | 0.762 | 0.764 | 0.763 | 0.766 | 0.767 | 0.768 | 0.768 | $0.765 \pm 0.001$ |
| | | **FATE** | **0.770** | **0.769** | **0.771** | **0.772** | **0.773** | **0.774** | **0.774** | **0.774** | $\mathbf{0.772} \pm 0.001$ |
| | DeepFM | Base | 0.772 | 0.771 | 0.772 | 0.772 | 0.774 | 0.774 | 0.774 | 0.774 | $0.773 \pm 0.001$ |
| | | Pooling | 0.772 | 0.772 | 0.773 | 0.774 | 0.776 | 0.776 | 0.776 | 0.776 | $0.774 \pm 0.002$ |
| | | **FATE** | **0.781** | **0.780** | **0.782** | **0.782** | **0.784** | **0.784** | **0.784** | **0.784** | $\mathbf{0.783} \pm 0.001$ |

*Base-NN*, *Average-NN*, *Pooling-NN* and *KNN-NN* are all trained on training instances with observed features. Then *Base-NN* only uses test instances' observed features for inference. *Average-NN* uses averaged embeddings of observed features as those of unobserved ones. *Pooling-NN* (resp. *KNN-NN*) computes embeddings for unobserved features via replacing our GNN with mean pooling aggregation over neighborhoods (resp. KNN aggregation over all the observed ones). Furthermore, we consider *Oracle-NN* using all the features of training data for training and *INL-NN* that is first trained on training data with observed features and then re-trained on training data with the remaining features.

**Results and Discussions.** Fig. 5 reports the mean Accuracy/ROC-AUC of five trials with different ratios of observed features ranging from 0.3 to 0.8. FATE achieves averagely 7.3% higher Accuracy and 1.3% higher ROC-AUC over Base-NN which uses partial features for inference. The improvements are statistically significant under 95% confidence level. The results show that FATE can learn effective embeddings for new features that contribute to better performance for classification. Furthermore, FATE achieves averagely 29.8% higher Accuracy and 0.5% higher ROC-AUC over baselines Average-NN, KNN-NN and Pooling-NN. These baseline methods perform worse than Base-NN especially when observed features are few, which suggests that directly aggregating embeddings of observed features for extrapolation would degrade the performance. By contrast, FATE possesses superior capability for extrapolating to new unseen features. Even with 30% observed features the model is able to distill the useful knowledge from 70% unobserved features without re-training, providing decent classification performance. Notably, compared with INL-NN, FATE even achieves higher accuracy in most cases with a 4.1% Accuracy improvement on average. The possible reason is that INL-NN is prone for over-fitting on new data and forgetting the previous one. Finally, FATE achieves very close performance to Oracle-NN when using sufficient observed features and can even slightly exceeds it with 80% features in Gene, Robot and Github. In fact, the GNN network in FATE can not only achieve feature extrapolation, but also capture feature-level relations, which could be another merit of our method.

## 4.2 Experiment on CTR Prediction

**Setup.** We split the dataset in chronological order to simulate real-world cases. For Avazu dataset which contains time information in ten days, we use the data of first/second day for training/validation and the data of the third to tenth days for test. For Criteo dataset whose records are given in temporal order, we split the dataset into ten continual subsets with equal size and use the first/second subset for training/validation and the third to tenth subsets for test. With such data splitting, we can naturally obtain validation/test data with a mixture of seen and unseen features in the training data (the new features come from new values out of the known range of raw features). For Avazu/Criteo, there are $\sim 0.6/\sim 1.3$ million features in training data, $\sim 0.2/\sim 0.4$ million new features in validation data and totally $\sim 1.1/\sim 0.8$ million new features in all the test splits. We use ROC-AUC as evaluation metric.

**Implementation Details.** We consider two specifications for our backbone: 1) a 3-layer feedfoward NN, and 2) DeepFM [17], a widely used model for advertisement click prediction considering inter-feature interactions over NN. The GNN is a 2-layer GraphSAGE model. The training method is inductive learning with k-shot samples and mini-batch partition. Also, we compare with baselines *Base* and *Pooling*. The *KNN* method would suffer from scalability issue in the two large datasets.

**Results and Discussions.** Table 1 reports the ROC-AUC results for different test splits, which show that FATE significantly outperforms Base and Pooling in all the test splits using NN and

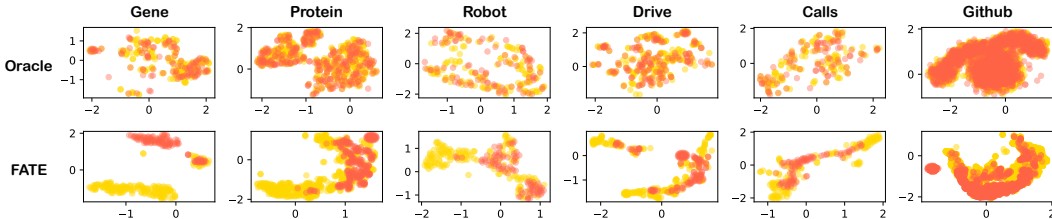

Figure 6: Visualization for t-SNE embeddings of FATE-NN's and Oracle-NN's produced feature embeddings. We mark observed features with red and the remaining ones with blue nodes.

DeepFM as backnones. Overall, compared with Base, FATE improves the ROC-AUC by 0.017/0.016 (resp. 0.008/0.01) with NN/DeepFM as the backbone on Avazu (resp. Criteo). Note that even an improvement of 0.004 for ROC-AUC is considered significant in click prediction tasks [13, 17]. The results show that FATE can combine useful information in new features collected in the future for the target task without further training and has promising power for enhancing the real-world systems interacting with open world. Compared with Pooling, FATE achieves significant AUC improvements on two datasets. The reason is that directly using average pooling to replace the GNN convolution would lead to limited capacity and weakens its ability for effective concept abstraction and reasoning.

### 4.3 Further Discussions

**Scalability.** We study model's scalability w.r.t. different batch sizes $B$ and number of features $D$ in Fig. 6 and 7 which show that our training and inference time/space scale linearly on Criteo dataset.

**Ablation Studies.** Table 3 also provides ablation studies, which show that 1) the DropEdge operation can help regularize the training and bring up higher test accuracy; 2) using asynchronous updates for two networks leads to performance gain over joint training; 3) the n-fold splitting and k-shot sampling will outperform each other in different cases and both exceeds leave-one-out partition. Table 4 compares using different sampling size $k$ for inductive learning approach on Avazu and Criteo, which verify our theoretical analysis in Section 3. See more discussions in Appendix F.

**Visualization.** Fig. 6 visualizes the produced feature embeddings by FATE-NN and Oracle-NN. It unveils two interesting insights that can interpret why FATE can sometimes outperform Oracle that uses all the features for training. First, FATE-NN's produced embeddings for observed and unobserved features possess more dissimilar distributions in latent space, compared to Oracle-NN. Notice that features' embeddings are used for the backbone network to compute intermediate hidden representation for each instance. Such phenomenon implies that FATE manages to extract more informative knowledge from new features. Second, the embeddings of FATE-NN form some particular structures (clusters, lines or curves) rather than uniformly distribute over the 2-D plane like Oracle-NN. The reason is that the GNN network leverages locality structures among features for further abstraction which explicitly encodes feature-level relations and could help downstream classification.

## 5 Connection to Other Learning Paradigms

Our introduced problem setting, open-world feature extrapolation (OFE), can be treated as an instantiation of out-of-distribution generalization [7, 33] or domain shift problem, focusing on distribution shift led by feature space expansion. We next discuss the relationships of our problem setting and our model FATE with domain adaption (DA), continual learning (CL), open-set learning (OSL) and zero-shot learning (ZSL). In general, OFE is orthogonal to these problems and opens a new direction that can potentially have promising intersections with the well-established ones.

**Domain Adaption** adapts a model trained on source domain to target with different distribution [4, 8, 32, 15, 16, 57, 31]. Our problem OFE is different from DA in two aspects: 1) the label space/distribution of training and test data is the same for OFE, while DA often mostly considers different label distributions for source and target domains; 2) OFE focus on combining new features that are related to the current task, while DA considers different tasks from different domains.

**Continual Learning**, or lifelong learning, aims at enabling a single model to learn from a stream of data from different tasks that cannot be seen at one time [39, 42, 36, 35]. By contrast, there are two-fold differences of OFE. First, OFE does not allow finetuning or further retraining on new data,

which can be more challenging than CL. Second, CL mostly assume each piece of data in the stream is from different tasks with different labels to handle. The key challenge of CL is the catastrophic forgetting [39, 35] that requires the model to balance a trade-off between previous and new tasks, while FATE is free from such issue in nature since we do not require incremental learning.

**Open-set Learning** is another line of researches that relate with us. Differently, open-set recognition mostly focus on expansion of label sets [5, 6, 21, 54, 34, 44]. To our best knowledge, we are the first to study feature set expansion, formulate it as OFE and further solve it via graph learning.

**Zero-shot Learning.** Our problem setting is also linked with few-shot/zero-shot learning. In NLP domains, some studies focus on dealing with rare entities exposed in limited times or new entities unseen by training [43, 30]. Similar problems are also encoutered and explored in cold-start recommender systems where there are also new users/items unseen before [51, 19, 50]. One common nature of these works aim at inferring the embeddings for new entities based on some 'held-out' ones. With a similar end and distinct methodological aspect, a recent study [50] explores a new possibility via learning a latent graph between existing entities (users) and newly arrived ones through attention mechanism for inductive representation learning. The core technical contributions of our work lie in the unique problem setting which stays focused on feature space expansion (domain shift) and the proposed feature-level sampling/partition training strategy, backed up with our theoretical insights.

**Extension and Outlooks.** Our work can be extended to solve more problems and push the development in broader areas. First, the input feature vectors can be replaced by feature maps given by CNN or word/sentence embeddings by Transformer [46], for handling multi-modal data in the context of federated learning [29] or multi-view learning [45, 2, 3]. Admittedly, our formulation assumes multi-hot feature vectors as input, which is a common practice for handling attribute features but not often the case for other data format (like vision or texts). For practitioners who would like to apply FATE to broader areas, one can extend our graph representation and GNN model to directly handle continuous features by treating feature values as edge weights as is done by [55]. Second, as shown in our experiments, the classification layers can be replaced by more complex models with inter-feature interactions [24, 28, 22, 27] or advanced architectures [49, 48] for more sufficient feature-wise interaction and improving the expressiveness.

## 6   Conclusion

We present a new framework to address new features unseen in training, by formulating it as the open-world feature extrapolation problem. We target the problem via graph representation learning by treating observed data as a feature-data graph and further harness GNN to inductively compute embeddings for new features with those of existing ones, mimicking abstraction and reasoning process in human's brain. We also propose two training strategies for effective feature extrapolation learning. Our theoretical results show that generalization error depends on training features and learning algorithms. Experiments verify its effectiveness and scalability to large-scale systems.

**Potential Societal Impacts.** When learning mapping from features to labels, the model is at risk of focusing on dominant features from majority groups and ignoring scarce features from minority ones. Potential extended works of much significance are to develop debiased methods for feature extrapolation. We believe AI models can be guided to promote social justice and well-being.

## Acknowledgments and Disclosure of Funding

This work was partly supported by National Key Research and Development Program of China (2020AAA0107600), Shanghai Municipal Science and Technology Major Project (2021SHZDZX0102), and NSFC (61972250, 72061127003).

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
