# A    More Discussions for Permutation-Invariant Property of Embedding Layers

In Section 2.1, we mentioned that the embedding layer in the backbone network can be equivalently seen as a combination of embedding lookup and a sum aggregation which is permutation-invariant w.r.t. the order of input features. We provide an illustration for this in Fig. 2.

**Equivalence between Concatenation and Sum Aggregation.** To support the remark argument in Section 2.1, we next illustrate the equivalence between concatenation of features' embeddings and sum aggregation/pooling over features' embeddings. Assume we have feature embeddings $\{\mathbf{z}_i^m\}_{m=1}^d$ for instance $\mathbf{x}_i$. We concat all the embeddings as a vector $\overline{\mathbf{z}_i} = [\mathbf{z}_i^1, \mathbf{z}_i^2, \cdots, \mathbf{z}_i^m]$ and feed it into a neural layer to obtain $\mathbf{c}_i = \mathbf{W}'\overline{\mathbf{z}_i}$. Notice that the weight matrix $\mathbf{W}' \in \mathbb{R}^{dH \times H}$ can be decomposed into $d$ sub-matrices $\{\mathbf{W}'_m\}_{m=1}^d$ where $\mathbf{W}'_m = \mathbf{W}'[(m-1)H : mH, :] \in \mathbb{R}^{H \times H}$. If we consider sum aggregation/pooling over $\{\mathbf{z}_i^m\}_{m=1}^d$, i.e. $\mathbf{z}_i = \sum_{m=1}^d \mathbf{z}_i^m$, the subsequent neural layer would be a weight matrix $\mathbf{W}''$ with dimension $H \times H$. We can set it as $\mathbf{W}'' = \sum_{m=1}^d \mathbf{W}'_m$ and will easily obtain $\mathbf{z}_i \mathbf{W}'' = \overline{\mathbf{z}_i}\mathbf{W}'$. Hence, the concatenation plus a fully-connected layer is equivalent to sum pooling plus a fully-connected layer. This observation indicates that our reasoning in the maintext can be applied to general neural network-based models for attribute features and enable them to handle input vectors with variable-length features.

# B    Training Algorithms

We present the training algorithms for our model in Alg. 1 where the model is trained end-to-end via self-supervised learning or inductive learning approaches.

# C    Analysis of Generalization Error

We provide a complete discussion and proof for analysis on generalization error of our approach. Some notations are repeatedly defined in order for a self-contained presentation in this section. Recall that we focus our analysis on the case of inductive learning with k-shot sampling approach. Also, we simplify the model as :1) the backbone network is a two-layer FFN (an embedding layer $\mathbf{W}$ plus a fully-connected layer $\Phi$) with sigmoid output; 2) the GNN network is a $L$-layer GCN which takes mean pooling aggregation over neighbored nodes without linear transformation and non-linearity in each layer; 3) the training algorithm is SGD.

**Derivation for model function.** With our settings in Section 3, we write the model as

$$\hat{y}_i = h(\mathbf{x}_i; \phi, \hat{\mathbf{W}}) = h(\mathbf{x}_i; \phi, g(\mathbf{W}, \mathbf{X}; \omega)) = \sigma\left(\sum_{i' \in \mathcal{N}_{\tilde{L}}(i) \cup \{i\}} c_{ii'}^L \mathbf{x}_{i'} \mathbf{W}\Phi\right), \tag{8}$$

where $\mathcal{N}_{\tilde{L}}(i)$ is a set which contains $\mathbf{x}_{i'}$'s that appear in the $\tilde{L}$-hop neighbors of $\mathbf{x}_i$ in the feature-data graph, $\tilde{L} = 2 \cdot \lfloor \frac{L}{2} \rfloor$, and $c_{ii'}^L$ is a weight that quantifies influence of $\mathbf{x}_i$ on $\mathbf{x}_{i'}$ through $L$-layer mean-pooling graph convolution. Here we provide the detailed derivation. In fact, the embedding layer in the backbone can be seen as a one-layer GCN convolution using sum pooling without linear transformation and non-linearity, which can be denoted by $\hat{Y} = \sigma(\mathbf{Z}\Phi) = \sigma(\mathbf{X}\hat{\mathbf{W}}\Phi)$ where $\mathbf{Z} = \{\mathbf{z}_i\}$ (recall that $\mathbf{X}$ is treated as an adjacency matrix of the feature-data graph in our model). The GNN model, which is a $L$-layer GCN with mean pooling without linear transformation and non-linearity in each layer, can be denoted by $\hat{\mathbf{W}} = (\mathbf{D}_{out}^{-1}\mathbf{X}^\top \mathbf{D}_{in}^{-1}\mathbf{X})^{\lfloor L/2 \rfloor}\mathbf{W}$, where $\mathbf{D}_{in} = diag(\{d_{in,i}\}_{i=1}^N)$ with $d_{in,i} = \sum_{j\in F} x_{ij}$ and $\mathbf{D}_{out} = diag(\{d_{out,j}\}_{j=1}^F)$ with $d_{out,j} = \sum_{i\in I} x_{ij}$. Hence, we have $\mathbf{Z} = \mathbf{X}(\mathbf{D}_{out}^{-1}\mathbf{X}^\top \mathbf{D}_{in}^{-1}\mathbf{X})^{\lfloor L/2 \rfloor}\mathbf{W} = (\mathbf{D}_{out}^{-1}\mathbf{D}_{in}^{-1})^{\lfloor L/2 \rfloor}\mathbf{X}(\mathbf{X}^\top \mathbf{X})^{\lfloor L/2 \rfloor}\mathbf{W} = (\mathbf{D}_{out}^{-1}\mathbf{D}_{in}^{-1})^{\lfloor L/2 \rfloor}(\mathbf{X}\mathbf{X}^\top)^{\lfloor L/2 \rfloor}\mathbf{X}\mathbf{W}$. Let $(\mathbf{D}_{out}^{-1}\mathbf{D}_{in}^{-1})^{\lfloor L/2 \rfloor}(\mathbf{X}\mathbf{X}^\top)^{\lfloor L/2 \rfloor} = \mathbf{C}_L \in \mathbb{R}^{N \times N}$ and $\mathbf{C}_L = \{c_{ii'}^L\}$ where $c_{ii'}^L$ denotes the a weight that quantifies influence between instance $i$ and $i'$ through $L$-layer GCN. Converting the global view of graph convolution into a local view for each node's ego-network, we can obtain $\hat{y}_i = \sigma(\mathbf{z}_i\Phi) = \sigma\left(\sum_{i' \in \mathcal{N}_{\tilde{L}}(i) \cup \{i\}} c_{ii'}^L \mathbf{x}_{i'}\mathbf{W}\Phi\right)$.

**Algorithm 1:** Training algorithm for feature extrapolation networks (FATE).

---

**1 INPUT:** $\mathbf{X}_{tr} = \{x_i\}_{i \in I_{tr}}$, training data matrix, $F_{tr} = \{f_j\}$, training feature set, $\omega^{(0)}$, initial parameter for GNN, $\theta^{(0)} = [\phi^{(0)}, \mathbf{W}^{(0)}]$, initial parameter for backbone network (where $\phi^{(0)}$ denotes parameter for classifer and $\mathbf{W}^{(0)}$ denotes initial feature embeddings), $n$ split ratio, $k$ sample size, $\rho$ dropedge ratio, $\alpha_s, \alpha_f$, learning rates.

**2 for** $t = 1, 2, \cdots, T_{max}$ **do**

**3**    Sample a mini-batch $\mathbf{X}^b = \{\mathbf{x}_i\}_{i \in I_b}$ from $\mathbf{X}_{tr} = \{\mathbf{x}_i\}_{i \in I_{tr}}$ // If handling large dataset, otherwise use $\mathbf{X}_{tr}$ directly ;

**4**    **if** *using self-supervised learning* **then**

**5**      Shuffle $F_{tr} = \{f_j\}$ and split into $n$ subsets $\{\overline{F}_s\}_{s=1}^n$ ;

**6**      **for** $s = 1, \cdots, n$ **do**

**7**        $\mathbf{W}^{(t)}[f_j] \leftarrow \mathbf{0}$, for $f_j \in \overline{F}_s$;

**8**        $\tilde{\mathbf{X}}^b = \text{DROPEDGE}(\mathbf{X}^b, \rho)$;

**9**        Feed $\mathbf{W}^{(t)}$ and $\tilde{\mathbf{X}}^b$ into GNN and obtain $\hat{\mathbf{W}}^{(t)} = g(\mathbf{W}^{(t)}, \tilde{\mathbf{X}}^b; \omega^{(t)})$;

**10**        $\hat{\mathbf{W}}^{(t)}[f_j] = \mathbf{W}^{(t)}[f_j]$, for $f_j \in \mathcal{F}_{tr} \setminus \overline{F}_s$;

**11**        Feed $\{\mathbf{x}_i\}_{i \in I_b}$ into backbone network and obtain $\{\hat{y}_i\}_{i \in I_b}$ by $\hat{y}_i = h(\mathbf{x}_i; \phi^{(t)}, \hat{\mathbf{W}}^{(t)})$;

**12**        Compute the loss $\mathcal{L}_s(Y^b, \hat{Y}^b) = \frac{1}{|I_{tr}|} \sum_{i \in Itr} l(\hat{y}_i, y_i)$;

**13**        Update $\theta^{(t+1)} \leftarrow \theta^t - \alpha_f \nabla_\theta \mathcal{L}_s(Y^b, \hat{Y}^b)$;

**14**      Update $\omega^{(t+1)} \leftarrow \omega^t - \alpha_s \sum_{s=1}^n \nabla_\omega \mathcal{L}_s(Y^b, \hat{Y}^b)$;

**15**    **if** *using inductive learning* **then**

**16**      **for** $s = 1, \cdots, n$ **do**

**17**        Sample $k$ raw features, extract a subset $F_s$ from $F_{tr} = \{f_j\}$ and extract $\mathbf{X}_s^b$ from $\mathbf{X}^b$ ;

**18**        $\tilde{\mathbf{X}}_s^b = \text{DROPEDGE}(\mathbf{X}_s^b, \rho)$;

**19**        Feed $\mathbf{W}^{(t)}$ and $\tilde{\mathbf{X}}_s^b$ into GNN and obtain $\hat{\mathbf{W}}^{(t)} = g(\mathbf{W}^{(t)}, \tilde{\mathbf{X}}_s^b; \omega^{(t)})$ ;

**20**        Feed $\{\mathbf{x}_i\}_{i \in I_b}$ into backbone network and obtain $\{\hat{y}_i\}_{i \in I_b}$ by $\hat{y}_i = h(\mathbf{x}_i; \phi^{(t)}, \hat{\mathbf{W}}^{(t)})$;

**21**        Compute the loss $\mathcal{L}_s(Y^b, \hat{Y}^b) = \frac{1}{|I_{tr}|} \sum_{i \in Itr} l(\hat{y}_i, y_i)$;

**22**        $\theta^{(t+1)} \leftarrow \theta^t - \alpha_f \nabla_\theta \mathcal{L}_s(Y^b, \hat{Y}^b)$;

**23**      Update $\omega^{(t+1)} \leftarrow \omega^t - \alpha_s \sum_{s=1}^n \nabla_\omega \mathcal{L}_s(Y^b, \hat{Y}^b)$;

**24 OUTPUT:** $\theta = [\phi, \mathbf{W}]$, trained parameter of backbone network, $\omega$, trained parameter of GNN.

---

With given training data $(\mathbf{X}_{tr}, Y_{tr})$, we define $\mathcal{S}$ as a set of all the data sub-matrices that could be sampled and exposed to the model during training

$$\mathcal{S} = \{(\mathbf{X}_1, Y_1), (\mathbf{X}_2, Y_2), \cdots, (\mathbf{X}_m, Y_m), \cdots, (\mathbf{X}_M, Y_M)\}.$$

The SGD training can be seen as a sequence of operations each of which picks an instance from $\mathcal{S}$ in an i.i.d. manner as a proxy training data and leverage it to compute updating gradient. We further introduce $\mathcal{S}^{\setminus m}$ which removes the $m$-th sub-matrix and $\mathcal{S}^m$ which replaces $m$-th sub-matrix by another one. Specifically, we have

$$\mathcal{S}^{\setminus m} = \{(\mathbf{X}_1, Y_1), \cdots, (\mathbf{X}_{m-1}, Y_{m-1}), (\mathbf{X}_{m+1}, Y_{m+1}) \cdots, (\mathbf{X}_M, Y_M)\},$$
$$\mathcal{S}^m = \{(\mathbf{X}_1, Y_1), \cdots, (\mathbf{X}_{m-1}, Y_{m-1}), (\mathbf{X}_{m'}, Y_{m'}), (\mathbf{X}_{m+1}, Y_{m+1}) \cdots, (\mathbf{X}_M, Y_M)\}.$$

**Justification of the i.i.d. Sampling.** In fact, for our inductive learning approach in Section 2.2, the observed features for each proxy data are randomly sampled. The feature-level sampling at one time can be seen as $k$ times i.i.d. sampling from all the raw features without replacement. Denote $C = \{c_n\}_{n=1}^d$ as a set of all the raw features in training set, $\mathcal{K}$ as a set of $k$ distinct indices in $\{1, \cdots, d\}$ and $C_\mathcal{K}$ denotes a subset of raw features with indices from $\mathcal{K}$. Obviously, there are $\binom{d}{k}$ different configurations for $\mathcal{K}$ (or $C_\mathcal{K}$) in total. We can equivalently treat once feature-level sampling as a one-time i.i.d. sampling from a set of candidates $\{\mathcal{K}_m\}$ which contains $\binom{d}{k}$ index sets and each index set $\mathcal{K}_m$ contains $k$ indices from $\{1, \cdots, d\}$. Next we discuss two cases.

1) If we do not consider instance-level mini-batch partition, then the set $\mathcal{S}$ will consist of $M = \binom{d}{k}$ sub-matrices. Specifically, the $m$-th sub-matrix $\mathbf{X}_m$ is induced by $C_{\mathcal{K}_m}$ which extracts the columns (corresponding to features generated by raw features in $C_{\mathcal{K}_m}$) of $\mathbf{X}_{tr}$.

2) If we use instance-level mini-batch partition, the case would be a bit more complicated. First of all, the instance-level partition is not a strictly i.i.d. sampling process over training instances since their exists dependency among different mini-batches in one epoch. Yet, in practice, the batch size $B$ is very large (e.g. $B = 100000$ in our experiment), so the number of mini-batches in one epoch is much smaller than $B$, which allows us to neglect the dependency in one epoch. Furthermore, since the instance-level selection is dependent of feature-level sampling, the whole sampling process for proxy data can be seen as a series of i.i.d. sampling over $\binom{N}{B} \times \binom{d}{k}$ sub-matrices of $\mathbf{X}_{tr}$, which consists of the set $\mathcal{S}$ in this case.

Next, we recall the generalization gap of our interests. The generalization error $R(h_{\mathcal{S}})$ is defined as

$$R(h_{\mathcal{S}}) = \mathbb{E}_{(\mathbf{X},Y)}[\mathcal{L}(Y, h(\mathbf{X}; \psi_{\mathcal{S}}))]. \tag{9}$$

where the expectation contains two stages of sampling: 1) a feature set $F = \{f_j\}$ is sampled according to $f_j \sim \mathcal{F}$, and 2) data $(\mathbf{X}, Y)$ is sampled according to $(\mathbf{x}_i, y_i) \sim \mathcal{D}_F$. The empirical risk that our approach optimizes with the training data would be

$$R_{emp}(h_{\mathcal{S}}) = \frac{1}{M} \sum_{m=1}^{M} \mathcal{L}(Y_m, h(\mathbf{X}_m; \psi_{\mathcal{S}})). \tag{10}$$

Then the expected generalization gap would be

$$\mathbb{E}_A[R(h_{\mathcal{S}}) - R_{emp}(h_{\mathcal{S}})], \tag{11}$$

where the expectation is taken over the randomness of $A_{\mathcal{S}}$ that stems from sampling in SGD.

We next prove the result in Theorem 1 in our maintext. Our proof is based on algorithmic stability analysis [9], following similar lines of reasoning in [20, 47]. The main idea of the stability analysis is to bound the output difference of a loss function from a single data point perturbation. Differently, in our case, the 'data point' is a data sub-matrix in $\mathcal{S}$. Therefore, our proof can be seen an extension of stability analysis to matrix data or graph as input. The proof can be divided into two parts. First, we derive a generalization error bound on condition of $\gamma$-uniform stability of the learning algorithm, Then we prove the bound for $\gamma$ based on our model architecture and SGD training.

### C.1  Generalization error with uniform stability condition

We first introduce the definition for uniform stability of a randomized learning algorithm as a building block of our proof. A randomized learning algorithm $A_{\mathcal{S}}$ is $\gamma$-uniform stable with regard to loss function $\mathcal{L}$ if it satisfies

$$\sup_{\mathcal{S}, (\mathbf{X}, Y)} |\mathbb{E}_{\mathcal{S}}[\mathcal{L}(Y, h(\mathbf{X}; \psi_{\mathcal{S}}))] - \mathbb{E}_S[\mathcal{L}(Y, h(\mathbf{X}; \psi_{\mathcal{S} \setminus m}))]| \leq \gamma. \tag{12}$$

We first prove a generalization bound using the uniform stability as a condition and then we prove that the learning algorithm in our case satisfies the condition.

**Theorem 2.** *Assume a randomized algorithm $A_{\mathcal{S}}$ is $\gamma$-uniform stable with a bounded loss function $0 \leq \mathcal{L}(Y, h(\mathbf{X}; \psi_{\mathcal{S}})) \leq L$. Then with probability at-least $1 - \delta$ ($0 < \delta < 1$), over the random draw of $\mathcal{S}$, we have*

$$\mathbb{E}_A[R(h_{\mathcal{S}}) - R_{emp}(h_{\mathcal{S}})] \leq 2 \cdot \gamma + (4M\gamma + L)\sqrt{\frac{\log \frac{1}{\delta}}{2M}}. \tag{13}$$

*Proof.* Using triangle inequality, the stability property in (12) yields,

$$
\begin{aligned}
&\sup_{\mathcal{S}, (\mathbf{X}, Y)} |\mathbb{E}_{\mathcal{S}}[\mathcal{L}(Y, h(\mathbf{X}; \psi_{\mathcal{S}}))] - \mathbb{E}_S[\mathcal{L}(Y, h(\mathbf{X}; \psi_{\mathcal{S}^m}))]| \\
&\leq \sup_{\mathcal{S}, (\mathbf{X}, Y)} |\mathbb{E}_{\mathcal{S}}[\mathcal{L}(Y, h(\mathbf{X}; \psi_{\mathcal{S}}))] - \mathbb{E}_S[\mathcal{L}(Y, h(\mathbf{X}; \psi_{\mathcal{S} \setminus m}))]| \\
&\quad + \sup_{\mathcal{S}, (\mathbf{X}, Y)} |\mathbb{E}_{\mathcal{S}}[\mathcal{L}(Y, h(\mathbf{X}; \psi_{\mathcal{S}^m}))] - \mathbb{E}_S[\mathcal{L}(Y, h(\mathbf{X}; \psi_{\mathcal{S} \setminus m}))]| \\
&\leq 2\gamma.
\end{aligned} \tag{14}
$$

We will use McDiarmid's concentration inequality for the following proof. Let $\mathbf{Z}$ be a random variable set and $f : \mathbf{Z}^M \to \mathbb{R}$. If it satisfies

$$\sup_{z_1,\cdots,z_m,\cdots,z_M,z'_m} |f(z_1,\cdots,z_m,\cdots,z_M) - f(z_1,\cdots,z'_m,\cdots,z_M)| \leq c_m, \qquad (15)$$

then we have

$$P(f(z_1,\cdots,z_M) - \mathbb{E}_{z_1,\cdots,z_M}[f(z_1,\cdots,z_M)] \geq \epsilon) \leq \exp\left(-\frac{2\epsilon^2}{\sum_{m=1}^M c_m^2}\right). \qquad (16)$$

Recall that data $(\mathbf{X}_s, Y_s)$ are assumed to be i.i.d. sampled, so we have (assuming $\mathbf{O} = (\mathbf{X}, Y)$)

$$\begin{aligned}
&\mathbb{E}_{\mathcal{S}}[\mathcal{L}(Y_m, h(\mathbf{X}_m; \psi_{\mathcal{S}}))] \\
&= \int \mathcal{L}(Y_m, h(\mathbf{X}_m; \psi_{\mathcal{S}}))p(\mathbf{O}_1,\cdots,\mathbf{O}_M)d\mathbf{O}_1\cdots d\mathbf{O}_M \\
&= \int \mathcal{L}(Y_m, h(\mathbf{X}_m; \psi_{\mathcal{S}}))p(\mathbf{O}_1)\cdots p(\mathbf{O}_M))d\mathbf{O}_1\cdots d\mathbf{O}_M \\
&= \int \mathcal{L}(Y_{m'}, h(\mathbf{X}_{m'}; \psi_{\mathcal{S}^m}))p(\mathbf{O}_1)\cdots p(\mathbf{O}_{m'})\cdots p(\mathbf{O}_M)d\mathbf{O}_1\cdots d\mathbf{O}_{m'}\cdots \mathbf{O}_M \\
&= \int \mathcal{L}(Y_m, h(\mathbf{X}_m; \psi_{\mathcal{S}^m}))p(\mathbf{O}_1,\cdots,\mathbf{O}_{m'},\cdots,\mathbf{O}_M)d\mathbf{O}_1\cdots d\mathbf{O}_{m'}\cdots d\mathbf{O}_M \times \int p(\mathbf{O}_m)d\mathbf{O}_m \\
&= \int \mathcal{L}(Y_{m'}, h(\mathbf{X}_{m'}; \psi_{\mathcal{S}^m}))p(\mathbf{O}_1,\cdots,\mathbf{O}_{m'},\mathbf{O}_m,\cdots,\mathbf{O}_M)d\mathbf{O}_1\cdots d\mathbf{O}_M d\mathbf{O}_{m'} \\
&= \mathbb{E}_{\mathcal{S},\mathbf{O}_{m'}}[\mathcal{L}(Y_{m'}, h(\mathbf{X}_{m'}; \psi_{\mathcal{S}^m}))].
\end{aligned} \qquad (17)$$

Using above equation and the $\gamma$-uniform stability we have

$$\begin{aligned}
&\mathbb{E}_{\mathcal{S}}[\mathbb{E}_A[R(h_{\mathcal{S}}) - R_{emp}(h_{\mathcal{S}})]] \\
&= \mathbb{E}_{\mathcal{S}}[\mathbb{E}_{\mathbf{O}}[\mathbb{E}_A[\mathcal{L}(Y, h(\mathbf{X}; \psi_{\mathcal{S}}))]]] - \frac{1}{M}\sum_{m=1}^M \mathbb{E}_{\mathcal{S}}[\mathbb{E}_A[\mathcal{L}(Y_m, h(\mathbf{X}_m; \psi_{\mathcal{S}}))]] \\
&= \mathbb{E}_{\mathcal{S}}[\mathbb{E}_{\mathbf{O}}[\mathbb{E}_A[\mathcal{L}(Y, h(\mathbf{X}; \psi_{\mathcal{S}}))]]] - \mathbb{E}_{\mathcal{S}}[\mathbb{E}_A[\mathcal{L}(Y_m, h(\mathbf{X}_m; \psi_{\mathcal{S}}))]] \\
&= \mathbb{E}_{\mathcal{S},\mathbf{O}_{m'}}[\mathbb{E}_A[\mathcal{L}(Y_{m'}, h(\mathbf{X}_{m'}; \psi_{\mathcal{S}}))]] - \mathbb{E}_{\mathcal{S},\mathbf{O}_{m'}}[\mathbb{E}_A[\mathcal{L}(Y_{m'}, h(\mathbf{X}_{m'}; \psi_{\mathcal{S}^m}))]] \\
&= \mathbb{E}_{\mathcal{S},\mathbf{O}_{m'}}[\mathbb{E}_A[\mathcal{L}(Y_{m'}, h(\mathbf{X}_{m'}; \psi_{\mathcal{S}})) - \mathcal{L}(Y_{m'}, h(\mathbf{X}_{m'}; \psi_{\mathcal{S}^m}))]] \\
&\leq \mathbb{E}_{\mathcal{S},\mathbf{O}_{m'}}[\mathbb{E}_A[|\mathcal{L}(Y_{m'}, h(\mathbf{X}_{m'}; \psi_{\mathcal{S}})) - \mathcal{L}(Y_{m'}, h(\mathbf{X}_{m'}; \psi_{\mathcal{S}^m}))|]] \\
&\leq 2\gamma.
\end{aligned} \qquad (18)$$

Also we have the following inequalities,

$$\begin{aligned}
|\mathbb{E}_A[R(h_{\mathcal{S}}) - R(h_{\mathcal{S}^m})]| &= |\mathbb{E}_{\mathbf{O}}[\mathbb{E}_A[\mathcal{L}(Y, h(\mathbf{X}; \psi_{\mathcal{S}}))]] - \mathbb{E}_{\mathbf{O}}[\mathbb{E}_A[\mathcal{L}(Y, h(\mathbf{X}; \psi_{\mathcal{S}^m}))]]| \\
&= |\mathbb{E}_{\mathbf{O}}[\mathbb{E}_A[\mathcal{L}(Y, h(\mathbf{X}; \psi_{\mathcal{S}}))]] - \mathbb{E}_A[\mathcal{L}(Y, h(\mathbf{X}; \psi_{\mathcal{S}^m}))]| \\
&\leq \mathbb{E}_{\mathbf{O}}[\mathbb{E}_A[|\mathcal{L}(Y, h(\mathbf{X}; \psi_{\mathcal{S}})) - \mathcal{L}(Y, h(\mathbf{X}; \psi_{\mathcal{S}^m}))|]] \\
&\leq 2\gamma,
\end{aligned} \qquad (19)$$

$$\begin{aligned}
|\mathbb{E}_A[R_{emp}(h_{\mathcal{S}}) - R_{emp}(h_{\mathcal{S}^m})]| \leq &|\frac{1}{M}\sum_{m'=1,m'\neq m}^M (\mathbb{E}_A[\mathcal{L}(Y_{m'}, h(\mathbf{X}_{m'}; \psi_{\mathcal{S}})) - \mathcal{L}(Y_{m'}, h(\mathbf{X}_{m'}; \psi_{\mathcal{S}^m}))])| \\
&+ |\frac{1}{M}\mathbb{E}_A[\mathcal{L}(Y_m, h(\mathbf{X}_m; \psi_{\mathcal{S}})) - \mathcal{L}(Y_{m'}, h(\mathbf{X}_{m'}; \psi_{\mathcal{S}^m}))]| \\
\leq &2\frac{M-1}{M}\gamma + \frac{\lambda}{M} \\
\leq &2\gamma + \frac{\lambda}{M}.
\end{aligned} \qquad (20)$$

Letting $K_\mathcal{S} = R(h_\mathcal{S}) - R_{emp}(h_\mathcal{S})$ and using (19) and (20), we obtain

$$
\begin{aligned}
|\mathbb{E}_A[K_\mathcal{S}] - \mathbb{E}_A[K_{\mathcal{S}^m}]| &= |\mathbb{E}_A[R(h_\mathcal{S}) - R_{emp}(h_\mathcal{S})] - \mathbb{E}_A[R(h_{\mathcal{S}^m}) - R_{emp}(h_{\mathcal{S}^m})]| \\
&\leq |\mathbb{E}_A[R(h_\mathcal{S})] - \mathbb{E}_A[R(h_{\mathcal{S}^m})]| + |\mathbb{E}_A[R_{emp}(h_\mathcal{S})] - \mathbb{E}_A[R_{emp}(h_{\mathcal{S}^m})]| \\
&\leq 2\gamma + (2\gamma + \frac{\lambda}{M}) \\
&\leq 4\gamma + \frac{\lambda}{M}.
\end{aligned}
\tag{21}
$$

Based on the above fact, we can apply the result of (16),

$$
P(\mathbb{E}_A[K_\mathcal{S}] - \mathbb{E}_\mathcal{S}[\mathbb{E}_A[K_\mathcal{S}]] \geq \epsilon) \leq \exp\left(-\frac{2\epsilon^2}{M(4\gamma + \frac{\lambda}{M})^2}\right).
\tag{22}
$$

Letting $\delta = \exp\left(-\frac{2\epsilon^2}{M(4\gamma + \frac{\lambda}{M})^2}\right)$ and using (18), we obtain the following result and conclude the proof.

$$
P\left(\mathbb{E}_A[K_\mathcal{S}] \leq 2\gamma + (4M\gamma + \lambda)\sqrt{\frac{\log(1/\delta)}{2M}}\right) \geq 1 - \delta.
\tag{23}
$$

$\square$

### C.2 Deriving bound for $\gamma$

We proceed to prove our main result in Theorem 1 by deriving the bound for $\gamma$ based on the SGD algorithm and our GNN model. Let $\Theta_\mathcal{S}$ and $\Theta_{\mathcal{S}^m}$ denote the weight matrix of the classifier in the backbone network. Recall that our model is $\hat{y}_i = h(\mathbf{x}_i; \psi) = \sigma(\sum_{i' \in \mathcal{N}_{\tilde{L}}(i) \cup \{i\}} c_{ii'}^L \mathbf{x}_{i'} \mathbf{W}\Phi)$. Hence, we have

$$
\begin{aligned}
&|\mathbb{E}_{SGD}[\mathcal{L}(Y, h(\mathbf{X}; \psi_\mathcal{S})) - \mathcal{L}(Y, h(\mathbf{X}; \psi_{\mathcal{S}^m}))]| \\
&= \left|\mathbb{E}_{SGD}\left[\frac{1}{N}\sum_{i \in I} l(y_i, h(\mathbf{x}_i; \psi_\mathcal{S})) - \frac{1}{N}\sum_{i \in I} l(y_i, h(\mathbf{x}_i; \psi_{\mathcal{S}^m}))\right]\right| \\
&\leq \frac{\beta}{N}\mathbb{E}_{SGD}\left[\sum_{i \in I}|h(\mathbf{x}_i; \psi_\mathcal{S}) - h(\mathbf{x}_i; \psi_{\mathcal{S}^m})|\right] \quad \text{(since } l(\cdot, \cdot) \text{ is } \beta\text{-Lipschitz)} \\
&= \frac{\beta}{N}\mathbb{E}_{SGD}\left[\sum_{i \in I}\left|\sigma\left(\sum_{i' \in \mathcal{N}_{\tilde{L}}(i) \cup \{i\}} c_{ii'}^L \mathbf{x}_{i'} \mathbf{W}_\mathcal{S}\Phi_\mathcal{S}\right) - \sigma\left(\sum_{i' \in \mathcal{N}_L(i) \cup \{i\}} c_{ii'}^L \mathbf{x}_{i'} \mathbf{W}_{\mathcal{S}^m}\Phi_{\mathcal{S}^m}\right)\right|\right] \\
&\leq \frac{\beta}{N}\mathbb{E}_{SGD}\left[\sum_{i \in I}\left|\sum_{i' \in \mathcal{N}_{\tilde{L}}(i) \cup \{i\}} c_{ii'}^L \mathbf{x}_{i'} \mathbf{W}_\mathcal{S}\Phi_\mathcal{S} - \sum_{i' \in \mathcal{N}_L(i) \cup \{i\}} c_{ii'}^L \mathbf{x}_{i'} \mathbf{W}_{\mathcal{S}^m}\Phi_{\mathcal{S}^m}\right|\right] \\
&\qquad \text{(due to the fact } |\sigma(x) - \sigma(y)| \leq |x - y|) \\
&\leq \frac{\beta}{N}\mathbb{E}_{SGD}\left[\sum_{i \in I}\left\|\sum_{i' \in \mathcal{N}_{\tilde{L}}(i) \cup \{i\}} c_{ii'}^L \mathbf{x}_{i'}\right\|_2 \cdot \|\mathbf{W}_\mathcal{S}\Phi_\mathcal{S} - \mathbf{W}_{\mathcal{S}^m}\Phi_{\mathcal{S}^m}\|_2\right] \\
&\leq \frac{\beta}{N}\sum_{i \in I}\left\|\sum_{i' \in \mathcal{N}_{\tilde{L}}(i) \cup \{i\}} c_{ii'}^L \mathbf{x}_{i'}\right\|_2 \mathbb{E}_{SGD}[\|\mathbf{W}_\mathcal{S}\Phi_\mathcal{S} - \mathbf{W}_{\mathcal{S}^m}\Phi_{\mathcal{S}^m}\|_2].
\end{aligned}
\tag{24}
$$

We need to bound the two terms in (24). First, notice that for $\forall \mathbf{x}_i$, it satisfies $\|\mathbf{x}_i\|_2 \leq \sqrt{d}$ and $\|x_{ij}\|_1 \leq d$ and the graph convolution with mean pooling induce the fact that $\|\sum_{i' \in \mathcal{N}_{\tilde{L}}(i) \cup \{i\}} c_{ii'}^L \mathbf{x}_{i'}\|_1 \leq d$. Using the inequality of arithmetic and geometric means, we have $\|\sum_{i' \in \mathcal{N}_{\tilde{L}}(i) \cup \{i\}} c_{ii'}^L \mathbf{x}_{i'}\|_2 \leq \sqrt{d}$.

We proceed to bound the second term by considering the randomness of SGD. We can define $\Psi_{\mathcal{S}} = \mathbf{W}_{\mathcal{S}}\Phi_{\mathcal{S}}$ as model parameters and we need to derive bound for $\mathbb{E}_{SGD}[\|\Psi_{\mathcal{S}} - \Psi_{\mathcal{S}^m}\|_2]$. Then define a sequence of model parameters $\{\Psi_{\mathcal{S},0}, \Psi_{\mathcal{S},1}, \cdots, \Psi_{\mathcal{S},T}\}$ where $\Psi_{\mathcal{S},t}$ denotes the model parameters learned by SGD on $\mathcal{S}$ with the updating in $t$-th step as

$$\Psi_{\mathcal{S},t+1} = \Psi_{\mathcal{S},t} - \alpha\nabla_\Psi\mathcal{L}(h(\mathbf{X}_t; \Psi_{\mathcal{S},t}), Y_t) = \Psi_{\mathcal{S},t} - \alpha\frac{1}{N_t}\sum_{i\in I_t}\nabla_\Psi l(h(\mathbf{x}_i; \Psi_{\mathcal{S},t}), y_i). \quad (25)$$

Similarly, $\{\Psi_{\mathcal{S}^m,0}, \Psi_{\mathcal{S}^m,1}, \cdots, \Psi_{\mathcal{S}^m,T}\}$ denotes a sequence of model parameters learned by SGD on $\mathcal{S}^m$. We then derive bound for $\Delta\Theta_t = \Psi_{\mathcal{S},t} - \Psi_{\mathcal{S}^m,t}$ by considering two cases.

First, at step $t$, SGD picks data $(\mathbf{X}_t, Y_t)$ and $t \neq m$, i.e., $(\mathbf{X}_t, Y_t)$ exists in both $\mathcal{S}$ and $\mathcal{S}^m$. This case will happen with probability $\frac{M-1}{M}$. The derivative of model output is

$$\frac{\partial h(\mathbf{x}_i; \Psi)}{\partial \Psi} = \sigma'\left(\sum_{i'\in\mathcal{N}_{\tilde{L}}(i)\cup\{i\}} c_{ii'}^L\mathbf{x}_{i'}\Psi\right) \cdot \sum_{i'\in\mathcal{N}_{\tilde{L}}(i)\cup\{i\}} c_{ii'}^L\mathbf{x}_{i'}. \quad (26)$$

Using the fact $|\sigma'(x) - \sigma'(y)| \leq |\sigma(x) - \sigma(y)| \leq |x - y|$, we have

$$\|\nabla_\Psi\mathcal{L}(h(\mathbf{X}_t; \Psi_{\mathcal{S},t}), Y_t) - \nabla_\Psi\mathcal{L}(h(\mathbf{X}_t; \Psi_{\mathcal{S}^m,t}), Y_t)\|_2$$

$$\leq\frac{1}{N_t}\sum_{i\in I_t}\|\nabla_\Psi l(h(\mathbf{x}_i; \Psi_{\mathcal{S},t}), y_i) - \nabla_\Psi l(h(\mathbf{x}_i; \Psi_{\mathcal{S}^m,t}), y_i)\|_2$$

$$\leq\frac{\beta'}{N_t}\sum_{i\in I_t}\|\nabla_\Psi h(\mathbf{x}_i; \Psi_{\mathcal{S},t}) - \nabla_\Psi h(\mathbf{x}_i; \Psi_{\mathcal{S}^m,t})\|_2$$

$$\leq\frac{\beta'}{N_t}\sum_{i\in I_t}\left\|\sigma'\left(\sum_{i'\in\mathcal{N}_{\tilde{L}}(i)\cup\{i\}} c_{ii'}^L\mathbf{x}_{i'}\Psi_{\mathcal{S},t}\right) \cdot \sum_{i'\in\mathcal{N}_{\tilde{L}}(i)\cup\{i\}} c_{ii'}^L\mathbf{x}_{i'} - \sigma'\left(\sum_{i'\in\mathcal{N}_{\tilde{L}}(i)\cup\{i\}} c_{ii'}^L\mathbf{x}_{i'}\Psi_{\mathcal{S}^m,t}\right) \cdot \sum_{i'\in\mathcal{N}_{\tilde{L}}(i)\cup\{i\}} c_{ii'}^L\mathbf{x}_{i'}\right\|_2$$

$$\leq\frac{\beta'}{N_t}\sum_{i\in I_t}\left\|\sum_{i'\in\mathcal{N}_{\tilde{L}}(i)\cup\{i\}} c_{ii'}^L\mathbf{x}_{i'}\right\|_2\left\|\sigma'\left(\sum_{i'\in\mathcal{N}_{\tilde{L}}(i)\cup\{i\}} c_{ii'}^L\mathbf{x}_{i'}\Psi_{\mathcal{S},t}\right) - \sigma'\left(\sum_{i'\in\mathcal{N}_{\tilde{L}}(i)\cup\{i\}} c_{ii'}^L\mathbf{x}_{i'}\Psi_{\mathcal{S}^m,t}\right)\right\|_2$$

$$\leq\frac{\beta'}{N_t}\sum_{i\in I_t}\sqrt{d}\left\|\sum_{i'\in\mathcal{N}_{\tilde{L}}(i)\cup\{i\}} c_{ii'}^L\mathbf{x}_{i'}\Psi_{\mathcal{S},t} - \sum_{i'\in\mathcal{N}_{\tilde{L}}(i)\cup\{i\}} c_{ii'}^L\mathbf{x}_{i'}\Psi_{\mathcal{S}^m,t}\right\|_2 \quad \text{(due to } |\sigma'(x) - \sigma'(y)| \leq |x - y|\text{)}$$

$$\leq\frac{\beta'}{N_t}\sum_{i\in I_t}\sqrt{d}\left\|\sum_{i'\in\mathcal{N}_{\tilde{L}}(i)\cup\{i\}} c_{ii'}^L\mathbf{x}_{i'}\right\|_2\|\Psi_{\mathcal{S},t} - \Psi_{\mathcal{S}^m,t}\|_2$$

$$\leq\beta'd\|\Delta\Psi_t\|_2. \quad (27)$$

Second, at step $t$, SGD picks $(\mathbf{X}_t, Y_t)$ and $t = m$, i.e., $(\mathbf{X}_t, Y_t)$ picked by the algorithm on $\mathcal{S}$ and $(\mathbf{X}_t', Y_t')$ picked by the algorithm on $\mathcal{S}^m$ are distinct. This case would happen with probability $\frac{1}{M}$.

We have

$$\|\nabla_\Psi \mathcal{L}(h(\mathbf{X}_t; \Psi_{\mathcal{S},t}), Y_t) - \nabla_\Psi \mathcal{L}(h(\mathbf{X}_t'; \Psi_{\mathcal{S}^m,t}), Y_t')\|_2$$

$$\leq \frac{1}{N_t} \sum_{i \in I_t, j = I_t'[i]} \|\nabla_\Psi l(h(\mathbf{x}_i; \Psi_{\mathcal{S},t}), y_i) - \nabla_\Psi l(h(\mathbf{x}_i'; \Psi_{\mathcal{S}^m,t}), y_i')\|_2$$

(since $N_t = N_t'$ and assume $I_t'[i]$ denotes the $i$-th entry in $I_t'$)

$$\leq \frac{\beta'}{N_t} \sum_{i \in I_t, j = I_t'[i]} \left\| \sigma'\left(\sum_{i' \in \mathcal{N}_{\tilde{L}}(i) \cup \{i\}} c_{ii'}^L \mathbf{x}_{i'} \Psi_{\mathcal{S},t}\right) \cdot \sum_{i' \in \mathcal{N}_{\tilde{L}}(i) \cup \{i\}} c_{ii'}^L \mathbf{x}_{i'} \right.$$

$$\left. -\sigma'\left(\sum_{i' \in \mathcal{N}_{\tilde{L}}(j) \cup \{j\}} c_{ji'}^L \mathbf{x}_{i'} \Psi_{\mathcal{S}^m,t}\right) \cdot \sum_{i' \in \mathcal{N}_{\tilde{L}}(j) \cup \{j\}} c_{ji'}^L \mathbf{x}_{i'} \right\|_2 \tag{28}$$

$$\leq \frac{\beta'}{N_t} \sum_{i \in I_t, j = I_t'[i]} \left\| \sigma'\left(\sum_{i' \in \mathcal{N}_{\tilde{L}}(j) \cup \{j\}} c_{ji'}^L \mathbf{x}_{i'} \Psi_{\mathcal{S},t}\right) \cdot \sum_{i' \in \mathcal{N}_{\tilde{L}}(j) \cup \{j\}} c_{ji'}^L \mathbf{x}_{i'} \right\|_2$$

$$+ \frac{\beta'}{N_t} \sum_{i \in I_t, j = I_t'[i]} \left\| \sigma'\left(\sum_{i' \in \mathcal{N}_{\tilde{L}}(j) \cup \{j\}} c_{ji'}^L \mathbf{x}_{i'} \Psi_{\mathcal{S}^m,t}\right) \cdot \sum_{i' \in \mathcal{N}_{\tilde{L}}(j) \cup \{j\}} c_{ji'}^L \mathbf{x}_{i'} \right\|_2$$

$$\leq 2\beta' \sqrt{d},$$

where the last inequality is due to $|\sigma'(x)| \leq 1$.

Combining (27) and (28) we have

$$\mathbb{E}_{SGD}[\|\Delta\Psi_{t+1}\|_2]$$

$$\leq \frac{M-1}{M} \mathbb{E}_{SGD}\left[\|(\Psi_{\mathcal{S},t} - \alpha\nabla_\Psi\mathcal{L}(h(\mathbf{X}_t; \Psi_{\mathcal{S},t}), Y_t)) - (\Psi_{\mathcal{S}^m,t} - \alpha\nabla_\Psi\mathcal{L}(h(\mathbf{X}_t; \Psi_{\mathcal{S}^m,t}), Y_t))\|_2\right]$$

$$+ \frac{1}{M} \mathbb{E}_{SGD}\left[\|(\Psi_{\mathcal{S},t} - \alpha\nabla_\Psi\mathcal{L}(h(\mathbf{X}_t; \Psi_{\mathcal{S},t}), Y_t)) - (\Psi_{\mathcal{S}^m,t} - \alpha\nabla_\Psi\mathcal{L}(h(\mathbf{X}_{F_t'}; \Psi_{\mathcal{S}^m,t}), Y_{F_t'}))\|_2\right]$$

$$\leq \mathbb{E}_{SGD}[\|\Delta\Psi_t\|_2] + (1 - \frac{1}{M})\alpha\mathbb{E}_{SGD}\left[\|\nabla_\Psi\mathcal{L}(h(\mathbf{X}_t; \Psi_{\mathcal{S},t}), Y_t) - \nabla_\Psi\mathcal{L}(h(\mathbf{X}_t; \Psi_{\mathcal{S}^m,t}), Y_t)\|_2\right]$$

$$+ \frac{1}{M}\alpha\mathbb{E}_{SGD}\left[\|\nabla_\Psi\mathcal{L}(h(\mathbf{X}_t; \Psi_{\mathcal{S},t}), Y_t) - \nabla_\Psi\mathcal{L}(h(\mathbf{X}_t'; \Psi_{\mathcal{S}^m,t}), Y_t')\|_2\right]$$

$$= \mathbb{E}_{SGD}[\|\Delta\Psi_t\|_2] + (1 - \frac{1}{M})\beta' d\mathbb{E}_{SGD}[\|\Delta\Psi_t\|_2] + \frac{2}{M}\beta'\sqrt{d}$$

$$\leq (1 + \beta' d\mathbb{E}_{SGD}[|\Delta\Psi_t|] + \frac{2}{M}\beta'\sqrt{d}. \tag{29}$$

The above inequality yields,

$$\mathbb{E}_{SGD}[|\Delta\Psi_T|] \leq \frac{2\beta'\sqrt{d}}{M} \sum_{t=1}^{T}(1 + \beta' d)^{t-1}. \tag{30}$$

Plugging the result into (24) we will obtain,

$$\gamma \leq \frac{2\beta\beta' d}{M} \sum_{t=1}^{T}(1 + \beta' d)^{t-1}. \tag{31}$$

We complete the proof for Theorem 1.

# D  Dataset Information

## D.1  Dataset Information

We present detailed information for our used datasets concerning the data collection, preprocessing and statistic information.

Table 2: Information for experiment datasets. The Github dataset directly provides preprocessed 0-1 features.

| Dataset | Domain | #Instances | #Raw Feat. | Cardinality | #0-1 Feat. | #Class |
|---------|--------|-----------|-----------|-------------|-----------|--------|
| Gene | Life | 3190 | 60 | 4∼6 | 287 | 3 |
| Protein | Life | 1080 | 80 | 2∼8 | 743 | 8 |
| Robot | Computer | 5456 | 24 | 9 | 237 | 4 |
| Drive | Computer | 58509 | 49 | 9 | 378 | 11 |
| Calls | Life | 7195 | 10 | 4∼10 | 219 | 10 |
| Github | Social | 37700 | - | ∼ | 4006 | 2 |
| Avazu | Ad. | 40,428,967 | 22 | 5∼1611749 | 2,018,025 | 2 |
| Criteo | Ad. | 45,840,617 | 39 | 5∼541311 | 2,647,481 | 2 |

**UCI datasets.** The six datasets are provided by UCI Machine Learning repository [1]. They are from different domains, including biology, engineering and social networks. Gene dataset contains 60 DNA sequence elements, and the task is to recognize exon/intron boundaries of DNA. Protein dataset [10] consists of the expression levels of 77 proteins/protein modifications, genotype, treatment type and behavior, and the task is to identify subsets of proteins that are discriminant between eight classes of mice. Robot dataset is collected as a robot navigates through a room following a wall with 24 ultrasound sensor readings, and the task is to predict the robot behavior. Drive dataset is extracted from electric current drive signals with 49 attributes, and the task is to identify 11 different classes with different conditions. Calls dataset was created by segmenting audio records belonging to 4 different families, 8 genus, and 10 species, and the task is to identify the class of species. Github dataset [41] is a large social network of GitHub developers with their location, repositories starred, employer and e-mail address, and the task is to predict whether the GitHub user is a web or a machine learning developer.

The six UCI datasets have diverse statistics. Overall, they contain thousands of instances and dozens of raw features with a mix up of categorial and continuous ones. The categorical raw features have cardinality ranged from 2 to 12. As mentioned in Section 2.1, the cardinality means the number of possible values for a discrete feature. For continuous features in each dataset (if exist), we first normalize the values into 0-mean and 1-standard-deviation distribution and then hash them into 10 buckets with evenly partition between the maximum and the minimum. Then each raw feature can be converted into one-hot representation. After converting all the features into binary ones we get up to hundreds of 0-1 features for each dataset. Table 2 summarizes the basic information for each dataset.

**CTR prediction datasets.** The two click-through rate (CTR) prediction datasets have millions of instances and dozens of raw features with diverse cardinality. The goal of CTR prediction task is to estimate the probability that a user will click on an advertisement with the user's profile features and the ad's content features. In specific, Criteo[4] is a widely used public benchmark dataset for developing CTR models, which includes 45 million users' click records, 13 continuous raw features and 26 categorical ones [5]. We follow [24, 23] and use log transformation to convert the continuous features into discrete ones. Avazu[6] is another publicly accessible dataset for CTR prediction, which contains users' mobile behaviors including whether a displayed mobile ad is clicked by a user or not. It has 40 millions users' click records, 23 categorical raw feature spanning from user/device features to ad attributes (all are encoded to remove user identity information). The cardinality of different raw features for these two datasets are very diverse, ranging from 5 to a million. The raw features with very large cardinality include some id features, e.g. device id, site id, app id, etc. For each dataset, we convert each raw feature into one-hot representations and obtain 0-1 features. For features appearing less than 4 times we group them as one feature. After preprocessing, we obtain nearly 2 million 0-1 features for Avazu and Criteo as shown in Table 2.

---

[4]http://labs.criteo.com/2014/02/kaggle-display-advertising-challenge-dataset/

[5]In computational advertisement community, the raw features (e.g. site category, device id, device type, app domain, etc.) are often called *fields*. We call them raw features in our paper to keep the notation self-contained.

[6]https://www.kaggle.com/c/avazu-ctr-prediction

### D.2 Dataset Splits

**UCI datasets.** For each of UCI datasets, we first randomly partition all the instances into training/validation/test sets according to the ratio of 6:2:2. Then we randomly select a certain ratio ($30\% \sim 80\%$ in our experiments) of features as observed features and use the remaining as unobserved ones. The model is trained with observed features of training instances, validated with observed features of validation instances and tested with all the features of test instances.

**CTR prediction datasets.** As illustrated in Section 4, for Avazu/Criteo we split all the instances into ten folds in chronological order. Then we use the first fold for training, second fold for validation, and third to tenth folds for test. In such way, the validation data and test data will naturally contain new features not appeared in training data. Here we provide more illustration about this. As mentioned above, Avazu dataset contains 23 categorial raw features and some of them have very large cardinality. For example, the cardinality of raw features *app id* and *device id* are 5481 and 381763, respectively. In practical systems, there will be new apps introduced and new devices observed by the system as time goes by, and they play as new values out of the known range of existing raw features, which consist of new 0-1 features that are not unseen by the model (as introduced in the beginning of Section 2.1). Since we chronologically divide the dataset into training/validation/test sets, the validation and test sets would both contain a mixture of features seen in training set and new features unseen in training. Concretely, for Avazu, there are totally 618411 features in training set, 248614 new features (unseen by training data) in validation set, and totally 1151000 new features (unseen by both training and validation sets) in all the test sets. For Criteo, there are totally 1340248 features in training set, 472023 new features (unseen by training data) in validation set, and totally 835210 new features (unseen by both training and validation sets) in all the test sets.

## E  Implementation Details

We present implementation details for our experiments for reproducibility. We implement our model as well as all the baselines with Python 3.8, Pytorch 1.7 and Pytorch Geometric 1.6. The experiments are all run on a RTX 2080Ti, except for our scalability test in Section 4.3 where we use a RTX 8000.

### E.1 Details for UCI experiments

**Architectures.** For experiments on UCI datasets, the network architecture for our backbone network is

- A three-layer neural network with hidden size 8 in each layer.
- The activation function is ReLU.
- The output layer is a softmax function for multi-class classification or sigmoid for two-class classification.

The architecture for our GNN network is

- A four-layer GCN [25] network with hidden size 8 in each layer.
- Adding self-loop and using normalization for graph convolution in each layer.
- No activation unit is used.

**Training Details.** We adopt self-supervised learning approach with n-fold splitting. Concretely, in each epoch, we feed the whole training data matrix into the model and randomly divide all the observed features into $n = 5$ disjoint sets $\{\overline{F}_s\}_{s=1}^n$. Then a nested optimization is considered: 1) we update the backbone network with $n$ steps where in the $s$-th step, we mask observed features in $\overline{F}_s$; 2) then we update the GNN network with one step using the accumulated loss of the $n$ steps. The training procedure will repeat the above process until a given budget of 200 epochs. Also, in each epoch, the validation loss is averaged over $n$-fold data where for the $s$-th fold the features in $\overline{F}_s$ are masked and the model will use the remaining observed features for prediction. Finally, we report the test accuracy achieved by the epoch that gives the minimum logloss on validation dataset.

**Hyperparameters.** Other hyper-parameters are searched with grid search on validation dataset. We use the same hyperparameter settings for six datasets, which indicates that our model is dataset agnostic in some senses. The settings and searching space are as follows:

- The learning rates $\alpha_f$, $\alpha_s$ are searched within $[0.1, 0.01, 0.001]$. We set $\alpha_f = 0.01$ and $\alpha_s = 0.001$.
- The ratio for DropEdge $\rho$ is searched within $[0.0, 0.2, 0.5]$. We set $\rho = 0.5$.
- The fold number for data partition $n$ is searched within $[2, 5, 10]$. We set $n = 5$.

**Baselines.** All the baselines are implemented with a three-layer neural network, the same as the backbone network in our model. The baselines are all trained with a given budget of 200 epochs, and we report the test accuracy achieved by the epoch that gives the minimum logloss on validation dataset. The difference of them lies in the ways for leveraging observed and unobserved (new) features in training and inference. The detailed information for baseline methods is as follows.

- *Base-NN.* Use observed features of training instances for model training, and observed features of validation/test instances for model validation/test.
- *Oracle-NN.* User all the features of training instances for model training, and all the features of validation/test instances for model validation/test.
- *INL-NN.* The training process contains two stages. In the first stage, we train the model with initialized parameters using observed features of training instances for 200 epochs and save the model at the epoch that gives the minimum logloss on validation dataset (with observed features). In the second stage, we load the saved model in the first stage, train it using unobserved features of training instances for 200 epochs and report the test accuracy (using all the features) achieved by the epoch that gives the minimum logloss on validation dataset (using all the features).
- *Average-NN.* Use observed features of training instances for model training. In test stage, we average the embeddings of observed features as the embeddings of unobserved features. Then the model would use all the features of test instances for inference (by using the trained embeddings of observed features and estimated embeddings of unobserved ones).
- *Pooling-NN.* Use observed features of training instances for model training. In test stage, we replace the GNN model in FATE with mean pooling over neighbored nodes. Specifically, the embeddings of unobserved features are obtained by non-parametric message passing using mean pooling over the feature-data bipartite graph.
- *KNN-NN.* Use observed features of training instances for model training. In test stage, we compute the Jaccard similarity scores between any pair of observed and unobserved features. Then for each unobserved feature, its embedding is obtained by taking average of the embeddings of the observed features with top 20% Jaccard similarities as the target unobserved feature.

## E.2 Details for Avazu/Criteo experiments

**Architectures.** For experiments on Criteo and Avazu datasets, we consider two specifications for the backbone network. First, we specify it as a feedforward NN, whose architecture is

- A three-layer neural network with hidden size 10-400-400-1.
- The activation function is ReLU unit except the last layer using sigmoid.
- We use BatchNorm and Dropout with probability 0.5 in each layer.

Second, we specify it as DeepFM network [17], which also contains an embedding layer and a subsequent classification layer. The embedding layer is an embedding lookup $\mathbf{W}$ which maps each nonzero index in $\mathbf{x}_i$ to an embedding, denoted as $\{\mathbf{z}_i^m\}$ where $\mathbf{z}_i^m$ denotes the embedding for the $m$-th raw feature of instance $i$. The subsequent classification layer can be denoted by

$$\hat{y}_i = \sum_{j=1}^{D} w_j \cdot x_{ij} + \text{FM}(\{\mathbf{z}_i^m\}) + \text{FNN}(\mathbf{z}_i), \tag{32}$$

where $\mathbf{z}_i = \sum_{m=1}^{d} \mathbf{z}_i^m$, FNN is a feedforward neural network and FM is a factorization machine which can be denoted as

$$\text{FM}(\{\mathbf{z}_i^m\}) = \sum_{m,m'} < \mathbf{z}_i^m, \mathbf{z}_i^{m'} > . \tag{33}$$

For our model FATE-DeepFM, we use the GNN model to compute feature embeddings $\hat{\mathbf{W}}$ based on which we use the input feature vector of an instance $\mathbf{x}_i$ to obtain $\{\mathbf{z}_i^m\}$ and $\mathbf{z}_i$ and then plug into the subsequent classification layer.

The architecture for our GNN network is

- A two-layer GraphSAGE [18] network with hidden size 10 in each layer.
- No activation unit.

**Training Details.** We adopt inductive learning approach with k-shot sampling for training our model. Furthermore, we use instance-level mini-batch partition to control space cost. Concretely, in each epoch, we first randomly shuffle all the training instances and partition them into mini-batches with size $B$. Then for each mini-batch, we consider a training iteration where the backbone network is updated with $n$ steps and the GNN model is updated with one step. For $s$-th step update for the backbone, we uniformly sample $k$ raw features from $d$ existing ones, obtain a new feature set $F_s$ induced by the sampled $k$ raw features, and extract the corresponding columns in the data matrix to form a proxy data, i.e., a $B \times |F_s|$ sub-matrix from $\mathbf{X}_{tr}$. We then use the proxy data matrix to update the backbone. After $n$-step updates for the backbone, we update the GNN model with the accumulated loss of the $n$ steps.

All the models including FATE and baselines are trained with a given budget of 100 epochs. For every 10 training iteration, we compute the validation loss and ROC-AUC on validation dataset. Finally, we report the test ROC-AUC achieved by the epoch that gives the highest ROC-AUC on validation dataset.

**Hyperparameters.** Other hyper-parameters are searched with grid search on validation dataset. The settings and searching space are as follows:

- The learning rates $\alpha_f$, $\alpha_s$ are searched within $[0.1, 0.01, 0.001, 0.0001, 0.00001]$. We set $\alpha_f = 0.0001$ and $\alpha_s = 0.0001$ for NN as backbone. For DeepFM as backbone, we set $\alpha_f = 0.0001$ and $\alpha_s = 0.0001$ on Criteo, and $\alpha_s = 0.00001$ on Avazu.
- The ratio for DropEdge $\rho$ is searched within $[0.0, 0.2, 0.5]$. We set $\rho = 0.5$.
- The batch size $B$ is searched within $[10000, 20000, 100000, 200000]$. We set $B = 100000$.
- The sampling size $k$ for data partition is searched within $[7, 11, 13, 17, 20]$ for Avazu and $[13, 16, 21, 24, 27]$ for Criteo. We set $k = 17$ for Avazu and $k = 24$ for Criteo.

## F More Experiment Results

We supplement more experiment results as further discussions of our method, including salability tests and ablation studies.

### F.1 Scalability Test

We conduct experiments for scalability test on Criteo dataset. The scalability experiment is deployed on a RTX 8000 GPU with 48GB memory (though our comparison experiments in Section 4.1 and 4.2 require less than 12GB memory for each trial).

**Impact of Batch Sizes.** We statistic the running time per mini-batch for training and inference on Criteo dataset in Fig. 6(a) and (b) where the batch size is changed from 1e5 to 1e6. The results are taken average over 20 mini-batches. As we can see, as the batch size increases, the training time and inference time both increase linearly, which depicts that our model has linear scalability w.r.t. the number of instances for each update and inference. Also, in Fig. 6(c) and (d), we present the GPU memory cost for training and inference on Criteo dataset. As we can see, the space cost of FATE also increases linearly with respect with batch sizes. Indeed, as discussed in Section 3.2, the time and

space complexity of FATE is $O(Bd)$ using mini-batch training where $d$ is relative small value (up to a hundred). Hence, the empirical results verify our analysis.

**Impact of Feature Numbers.** We also discuss the model's scalability concerning different feature numbers, i.e. the dimension of feature vectors $D$ for training data ($D'$ for test data). There are totally 39 raw features in Criteo dataset and we only use $[39, 36, 33, 30, 27, 24, 21, 18, 15]$ of them for experiments, which induces $[2647481, 2475154, 1956637, 1949494, 1466711, 927535, 862927, 850016]$ features, and also compare the training/inference time per mini-batch and GPU memory costs. The results are shown in Fig. 7(a)-(d). We can see that as the feature number increases, the time and space costs both go up in linear trends, which indicates FATE has linear scalability w.r.t. feature number $D$. In fact, more feature numbers would require larger model size (for feature embeddings) and induce larger computational graph due to the increase of $d$; also, the increase of $d$ would also require more training/inference time based on our complexity analysis.

### F.2 Ablation Studies

We next conduct ablation studies for some key components in our framework and discuss their impacts on our model. The results are shown in Table 3 and Table 4.

**Effectiveness of DropEdge.** In Table 3, we compare with not using DropEdge regularization in training stage. The results show that FATE consistently achieve superior accuracy throughout six datasets, which demonstrate the effectiveness of DropEdge regularization that can help to alleviate the over-fitting on training features.

**Effectiveness of Asynchronous Updates.** We also compare our asynchronous updates (alternative fast updates for Backbone network and slow updates for GNN) with directly using end-to-end jointly training of two networks. The results show that FATE with asynchronous updates can outperform joint training approach over a large margin, which verify the effectiveness of our proposed asynchronous updating rule. The reason is that using asynchronous updates can decouple the training for two networks and further help two models learn useful information from observed data. Also, we observe that using slow updates for GNN network with the accumulated loss of several data splits can stabilize its training and alleviate the over-fitting.

**Comparison between Training Approaches.** We further investigate the k-fold splitting and n-fold sampling strategies used in our training approaches in Table 3. Recall that in UCI datasets, we adopt the self-supervised learning approach for training. Here we compare our used n-fold splitting with leave-one-out, which leave out partial features as a fixed set for masking in training, and k-shot sampling, which randomly sample $\lfloor 0.8 * D \rfloor$ training features as observed ones and mask the remaining for each update. The results show that the n-fold splitting and k-shot sampling strategies both provide superior performance in six datasets. Furthermore, when using different $n$'s, the relative performance of n-fold splitting and k-shot sampling approaches diverge in different cases. Overall, we found using n-fold splitting with $n = 5$ or $n = 10$ work the best on average. In fact, the n-fold splitting and k-shot sampling both play as a role in mimicking new features and exposing partial observed features to the model in training. The difference is that n-fold splitting guarantees that in each iteration the model can be updated on each feature in training set while the k-shot sampling introduces more randomness. Unlike UCI datasets, in two large-scale datasets Criteo and Avazu where we adopt the inductive learning approach for training, we found using k-shot sampling works consistently better than n-fold splitting. One possible reason is that k-shot sampling can increase the diversity of proxy data (containing partial features and partial instances) used for each training update and can presumably help the model to overcome feature-level over-fitting in large datasets. Such results are consistent with our theoretical generalization error analysis in Section 3.

**Impact of Sampling Sizes.** We next study the impact of sampling size $k$ on the model performance. We use different $k$'s for inductive learning on Avazu and Criteo. The results are shown in Table 4. As we can see, as $k$ increases, the training AUC goes up, which demonstrates that larger sampling size can help for optimization since it reduces the variance of sampling and enhances training stability. Furthermore, it is not always beneficial to increase $k$. When it becomes large enough and close to the number of raw features $d$, the model would suffers from over-fitting. The results further demonstrate that large sampling size would lead to feature-level over-fitting, which echoes our theoretical results in Section 3. Recall that Theorem 1 shows that model's generalization gap depends on the randomness in

sampling over training features. Here when $k$ is large, there will be less randomness from feature-level data partition, which will degrade model's generalization ability.

Table 3: Ablation studies for DropEdge regularization, asynchronous updates for two networks (compared with end-to-end joint training) and our sampling strategies (n-fold splits and k-shot sampling compared with leave-one-out) on six UCI datasets. We run each experiment five times with different random seeds and report the mean scores.

| Models | Gene | Protein | Robot | Drive | Calls | Github |
|---|---|---|---|---|---|---|
| w/o DropEdge | 0.9226 | 0.9031 | 0.8062 | 0.5261 | 0.9760 | 0.8688 |
| End-to-end Joint | 0.9257 | 0.8963 | 0.8454 | 0.1073 | 0.9762 | 0.7557 |
| **FATE (ours)** | **0.9345** | **0.9178** | **0.8815** | **0.6440** | **0.9839** | **0.8743** |
| Leave-one-out | 0.8564 | 0.6574 | 0.7641 | 0.4448 | 0.9334 | 0.8533 |
| n-fold split ($n = 10$) | 0.8884 | 0.8426 | **0.8888** | 0.5910 | **0.9851** | 0.8723 |
| **n-fold split** ($n = 5$) | 0.9345 | **0.9178** | 0.8815 | **0.6440** | 0.9839 | 0.8743 |
| n-fold split ($n = 2$) | 0.9298 | 0.8398 | 0.8359 | 0.5234 | 0.9514 | **0.8771** |
| k-shot sample ($n = 10$) | **0.9404** | 0.9046 | 0.8839 | 0.5559 | 0.9812 | 0.8712 |
| k-shot sample ($n = 5$) | 0.9379 | 0.9102 | 0.8802 | 0.6060 | 0.9819 | 0.8712 |
| k-shot sample ($n = 1$) | 0.9304 | 0.8778 | 0.8568 | 0.5408 | 0.9611 | 0.8722 |

Table 4: Ablation studies for different sampling sizes $k$ for k-shot sampling in inductive learning on Avazu and Criteo. We report ROC-AUC on training data, validation data and 8-fold test data (T1-T8).

| Dataset | $k$ | Train | Val | T1 | T2 | T3 | T4 | T5 | T6 | T7 | T8 |
|---|---|---|---|---|---|---|---|---|---|---|---|
| Avazu | 11 | 0.7815 | 0.7369 | 0.6853 | 0.6950 | 0.7058 | 0.7093 | 0.7137 | 0.7186 | 0.7183 | 0.7193 |
|  | 14 | 0.7842 | 0.7399 | 0.6896 | 0.6989 | 0.7080 | 0.7091 | 0.7142 | 0.7190 | 0.7201 | 0.7210 |
|  | 17 | 0.7902 | 0.7433 | 0.6894 | 0.6995 | 0.7082 | 0.7105 | 0.7156 | 0.7203 | 0.7215 | 0.7216 |
|  | 20 | 0.7978 | 0.7420 | 0.6872 | 0.6978 | 0.7080 | 0.7091 | 0.7146 | 0.7201 | 0.7201 | 0.7202 |
| Criteo | 16 | 0.7955 | 0.7725 | 0.7669 | 0.7666 | 0.7688 | 0.7695 | 0.7714 | 0.7721 | 0.7722 | 0.7722 |
|  | 21 | 0.7988 | 0.7752 | 0.7699 | 0.7695 | 0.7714 | 0.7721 | 0.7736 | 0.7739 | 0.7741 | 0.7744 |
|  | 24 | 0.8005 | 0.7758 | 0.7701 | 0.7694 | 0.7712 | 0.7727 | 0.7732 | 0.7745 | 0.7740 | 0.7743 |
|  | 27 | 0.8025 | 0.7747 | 0.7698 | 0.7683 | 0.7711 | 0.7713 | 0.7727 | 0.7743 | 0.7734 | 0.7744 |
|  | 30 | 0.8057 | 0.7750 | 0.7690 | 0.7678 | 0.7701 | 0.7708 | 0.7725 | 0.7735 | 0.7723 | 0.7739 |

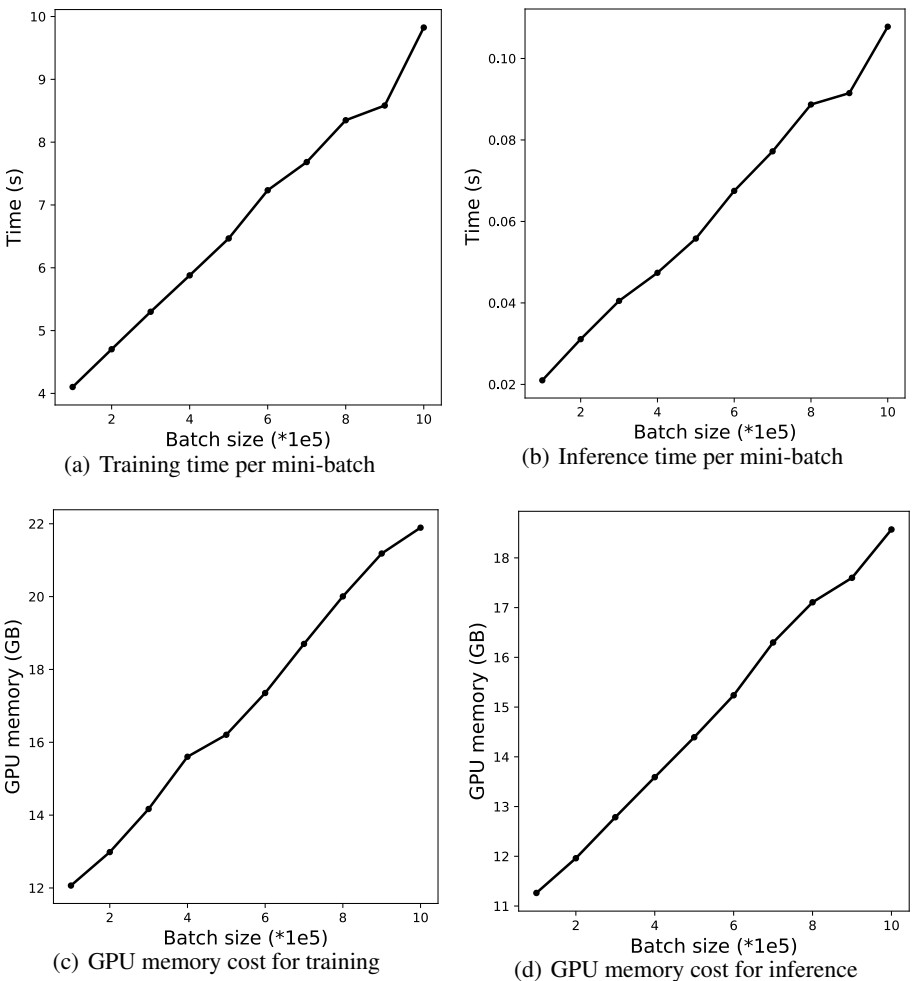

(a) Training time per mini-batch

(b) Inference time per mini-batch

(c) GPU memory cost for training

(d) GPU memory cost for inference

Figure 6: Scalability test of time and space costs w.r.t. batch sizes $B$ for training and inference on Criteo dataset.

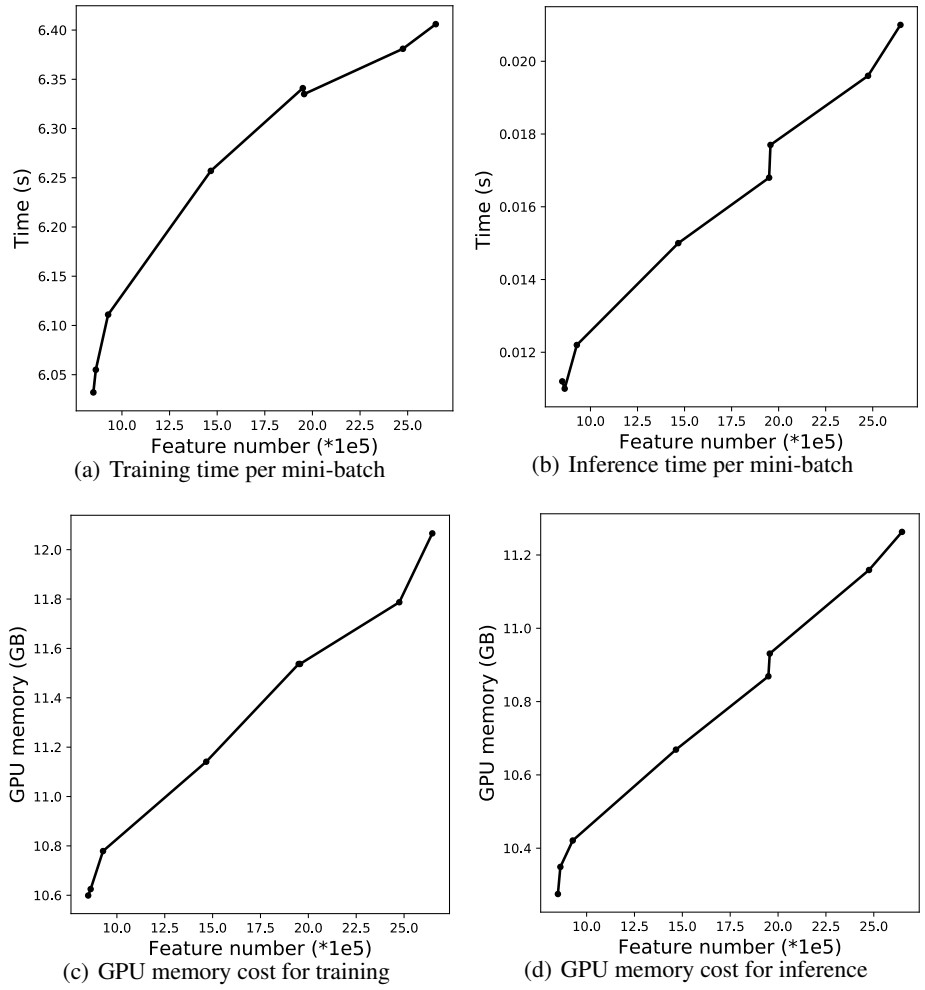

(a) Training time per mini-batch

(b) Inference time per mini-batch

(c) GPU memory cost for training

(d) GPU memory cost for inference

Figure 7: Scalability test of time and space costs w.r.t. feature numbers $D$ for training and inference on Criteo dataset.