# OpenReview forum: "Towards Open-World Feature Extrapolation: An Inductive Graph Learning Approach"
_NeurIPS.cc/2021/Conference — NeurIPS 2021 Poster_

### Official Review · Reviewer_T7WN · 2021-07-14

**Rating:** 7
**Confidence:** 4

**Summary:**

This paper tackles the unseen features with their extrapolation method from seen features, which is an important problem in machine learning.
* The authors first formulate the problem of open-world feature extrapolation that aims to deal with features unobserved at training while observed at inference.
* The authors propose to generate embedding for new features with inductive graph neural networks, where the graph-structured is constructed from features and instances of the observed training data.
* The authors theoretically analyze the generalization gap of the proposed method, and show that the number of features and the size of subsets for training data are important ingredients.

**Limitations And Societal Impact:**

The authors clearly outline the limitation and potential societal impact of their work in the conclusion section, and I'm satisfied with them.

**Main Review:**

### Strengths
* The problem of open-world feature extrapolation the authors introduce is a novel, interesting, and important direction for practical machine learning systems, since many new features are emerging in the real-world.
* The authors solve the proposed problem with a graph neural network (GNN) that can consider relevant seen features for extrapolating unseen features during neighborhood aggregation steps, which is convincing for tackling unseen. To be more specific, the authors divide the matrix multiplication from input features to latent features into two steps, in which I enjoy the concept of feature embedding with its replacing method using GNNs.
* The authors make an effort to analyze the generalization behavior of the proposed method, which is quite related to the empirical results.
* The proposed method shows performance improvements from baselines on two different datasets.
* The paper is well written, and the figures support the contents of the main paper well.

### Weaknesses
* The concept of extrapolating knowledge from seen to unseen with GNNs is not new, and there are few relevant work that the authors need to discuss (See Related Work below for details).

### Related Work
* These two relevant works [1, 2] should be discussed.
  * The first work [1] aims to embed unseen knowledge with inductive and transductive GNNs, while using meta-learning to simulate the unseen during training, compared to the authors' training schemes that divide or sample the whole observed training set to make the unseen.
  * The second work [2] aims to represent missing values in features with graph neural networks, while not tackling disjoint feature sets from training as the authors' one though.

### Clarity
* This paper is overally well written, however, there is an unclear part: how can we use the updated feature embeddings from GNNs $\hat{W}$ in the embedding layer. When I read the text in line 155, I think all the embeddings from GNN $\hat{W}$ are used as parameters for the embedding layer. However, the sentence in line 172 describes that the only used parameters for the embedding layer are the updated embeddings of masked features, not of all features. However, in line 181, I feel that the inductive learning scheme uses all updated embeddings, which is different from the description of line 172.

### Suggestions (or Questions)
* Most of the results in section 4.2 are in the supplementary file. It is better to state the location of the results, when the reference letters (e.g., Fig. 6) are given. At first glance, I think that the Figure 6 is in the main paper.
* In line 88, is there any reason to convert a continuous feature to a discrete one? I think this scheme results in the loss of information.

### Typo
* In the caption of Figure 1, (b) Oue approach.
* In Figure 5, there is no blue color on the visualization example, while the caption mentions blue.

---

[1] Baek et al. Learning to Extrapolate Knowledge: Transductive Few-shot Out-of-Graph Link Prediction. NeurIPS 2020.

[2] You et al. Handling Missing Data with Graph Representation Learning. NeurIPS 2020.

**Time Spent Reviewing:**

6 hours

---

> ### Author Response · Authors · 2021-08-09
> **Response to Reviewer T7WN**
>
> We thank the reviewer for the thorough review. The reviewer appreciates our problem setting as a novel and significant topic and also highlights our reasonable solution, related theoretical analysis, promising empirical results, and clear presentation. Below we answer the questions.
>
> **RE**: Differences to related works regarding knowledge extrapolation.
>
> The reviewer asks us to compare with two recent works [1] and [2] that associate with our method in different aspects. While these works consider using GNN's message passing for handling unseen nodes [1] and missing values [2], our framework focus on extrapolating knowledge in existing features (seen in training) to new features (unseen in training) in input space. The feature space goes through expansion and leads to distribution shift from training to test data, instead of staying static in [1, 2]. We next provide a detailed comparison and will incorporate this discussion in the final version.
>
> The work [1] deals with entity/link prediction in knowledge graph which may evolve with new unseen entities and adopts GNN for leveraging the known information from neighbored nodes/edges to estimate the representations of unseen ones. Essentially, such a problem can be treated as handling unseen nodes/edges on graphs, which is quite different from us. Concretely, the unseen nodes tackled by [1] can be treated as instances (which may have attribute features or not), while in our case we deal with unseen features (which consist of input components for each instance). Formally speaking, the distribution of input instances stays the same in [1] and its core challenge is to tackle out-of-graph nodes/edges. In our case, there is a distribution shift (given by augmentation of feature space) between training and testing instances, which requires us to handle out-of-distribution generalization.
>
> The work [2] focus on imputing missing values in tabular data and propose to convert missing-data imputation problem into a graph representation learning by treating observed data matrix as a feature-instance graph. While some feature values of partial instances are missing in dataset, the feature space is shared by training and test data. By contrast, we focus on extraploation to new features from augmented space, which requires the model for generalizing to new observation in-the-wild.
>
> **RE**: Clarification for updated embeddings used for subsequent decision-making layer.
>
> Thank you for pointing out the unclear part, which helps us to improve the presentation in the final version. Actually, the feature embeddings produced by GNN are used in different manners in our two proposed approach. As illustrated in Section 2.2, for the inductive learning approach, we use the updated embeddings by GNN for all the sampled features for subsequent layer. For the self-supervised approach, we use the updated embeddings by GNN for masked features and the initial embeddings (that do not go through GNN) for observed features, which we found works better in practice than using the updated embeddings for both sets of features.
>
> **RE**: Discretization of continuous features.
>
> This is a good question about information loss in discretization. In fact, converting continuous features to discrete ones is a common practice that is widely used to handle attribute features [3,4], especially used as important components in many data challenges like Kaggle. The reason is that compared with directly using continuous features as input, the preprocessed discrete features enable the model to learn embedding for each value which provides better representation capacity and expressiveness together with subsequent neural network and feature interaction layer. The improvement by the capacity is often more significant than the precision loss in discretization when using proper transformation. For example, in our exploratory experiments, we found using discretization for continuous features would often yield better performance than directly using the raw features when the transformation is properly considered. There are also solutions to controlling the information loss, such as using adaptive partitions/transformations for different features or advanced encoding/hashing scheme for preserving the information.
>
> **Minors.** Thanks for pointing out the typos and the useful suggestions. Indeed, in Fig. 5, the 'blue' should be 'yellow'. We will modify these minor issues in the final version.
>
>  References:
>
> [1] Learning to Extrapolate Knowledge: Transductive Few-shot Out-of-graph Link Prediction, in NeurIPS 2020.
>
> [2] Handling Missing Data with Graph Representation Learning, in NeurIPS 2020.
>
> [3] Field-aware Factorization Machines for {CTR} Prediction, in RecSys 2016.
>
> [4] Filed-wise Learning for Multi-filed Categorical Data, in NeurIPS 2020.

---

> > ### Comment · Reviewer_T7WN · 2021-08-17
> > **Thank you for your response**
> >
> > I really appreciate your comments. The details are below:
> >
> > * For differences to related works, the discussions of the proposed method with two recent works [1, 2] seem reasonable and valuable. I agree that [1] tackles the extrapolation from seen to unseen nodes, while the proposed work tackles the extrapolation to new features, thus the focus of work is indeed different despite considering GNNs for extrapolation on both methods. Also, while [2] uses the same feature space for training and test, the authors' setting is different and more challenging which uses different feature space across training and test. I hope that the authors include their discussions in the subsequent revision.
> >
> > * For clarification of updated embeddings used for the subsequent decision-making layer, I now understand that there are two ways to use feature embeddings from GNNs. Thank you for clarifying this with detailed explanations.
> >
> > * For the question of discretization of continuous features, the authors follow the conventional rule for handling attribute features, and there is no further concern on this.
> >
> > Thank you for making much effort to address my comments.

---

> > > ### Author Response · Authors · 2021-08-23
> > > **Thank you for the feedbacks**
> > >
> > > We thank the reviewer for the time and valuable feedbacks. We will incorporate the discussions on related works in our final version.

---

### Official Review · Reviewer_2Jqz · 2021-07-16

**Rating:** 7
**Confidence:** 3

**Summary:**

This paper formulates the feature extrapolation problem and proposes a novel framework to tackle it. By definition in the paper, the feature extrapolation problem emerges when, during test time, feature space extends beyond training set either due to the collection of new raw features or feature values going beyond its range. To handle such problems without requiring retraining a model in the occurrence of new features, the authors proposes to jointly train the backbone network, which performs the main classification task, and a GNN, which serves as a middleman to map the raw features to the input of the backbone network. This GNN is trained to find the co-occurrence relationships among feature values and extrapolate the representation for new features. Theoretical analysis was conducted to study the generalization error of the learning algorithm. Empirical study was carried out to demonstrate the effectiveness of the framework in solving tasks from real-world application.

**Limitations And Societal Impact:**

As stated in the paper, this work focuses on handling attribute features. The proposed method does not extend to problems in domains such as NLP and computer vision straightforwardly.

**Main Review:**

Overall, the paper is very well written and easy to follow. Formulation of the problem, proposed model and learning procedure are all clearly stated with minimal ambiguity. The proposed method is a reasonable solution to the problem. Empirical study shows a clear picture that the proposed method can utilize new features in test time without retraining of the backbone network.

The feature extrapolation problem formulated in this paper is quite interesting. The proposed method can handle two ways that feature space of test data extends beyond training data (1) there is a new feature available on test data; and (2) the range of values of an existing feature extend beyond its range on training data. Both cases may emerge in reality and the proposed method is useful to address such problems.

It seems that bin size used to quantizing continuous features plays a role in deciding the sparsity of the feature adjacency matrix which further affects the message passing on GNN. A finer-grained quantization preserves more information but leads to a sparser adjacency matrix. The paper can be strengthened by discussing such trade-off and its impact on the performance.

Permutation invariant aggregation provides the advantage that the final classifier network may consume feature vectors of variable lengths. Yet, this position information is lost during this transformation. In fact, when raw features are discretized and converted to embeddings, we also lost the knowledge about whether two feature embeddings come from the same raw feature.

Figure 5: the caption says “​​We mark observed features with red and the remaining ones with blue nodes.” There’s no blue nodes. It's probably a typo.

It would be great to add error bars in Figure 4 to show the significance of difference between different methods.








**Time Spent Reviewing:**

5

---

> ### Author Response · Authors · 2021-08-10
> **Response to Reviewer 2Jqz**
>
> We are glad that the reviewer liked our approach and appreciated that our paper is well written and motivated. The reviewer also considered our solution novel and reasonable, the experiment results convincing, and the problem of much practical significance. Below we provide responses to the raised (minor) issues.
>
> **RE**: The impact of bin size used to quantizing continuous features
>
> This is a good point and thank you for the suggestion. We agree that the bin size for quantizing continuous features would imapct the performance. Yet, this impact would also make differences to baselines that do not consider feature extrapolation. We ran new experiment using bin number 5, 10 and 15 for continuous features in dataset Proteins and Drive.
>
> - In Proteins dataset we get
> -- FATE: 0.9021 (5), 0.9009 (10), 0.8928 (15)
> -- Pooling-NN: 0.6389 (5), 0.6379 (10), 0.6276 (15)
> -- Oracle-NN: 0.9498 (5), 0.9459 (10), 0.9391 (15)
>
> - In Drive dataset we get
> -- FATE: 0.6597 (5), 0.6704 (10), 0.6690 (15)
> -- Pooling-NN: 0.5051 (5), 0.5114 (10), 0.5101 (15)
> -- Oracle-NN: 0.6803 (5), 0.6921 (10), 0.6905 (15)
>
> As you can see,  in Proteins dataset, there is no obvious difference between using bin number as 5 and 10 and the models' performance suffers a slight drop when it increases further. In Drive dataset, the accuracies of all the models first increase and then decrease as the bin number goes up. Indeed the results indicate that there exists a trade-off between sparsity and granularity in the discretization. One can notice that the Oracle-NN also exhibits similar trends as FATE and Pooling-NN. The reason is that the quality (e.g., sparsity and granularity) of preprocessed features plays a dominant role for embedding learning. Yet our exploratory experiments show that using discretization for continuous features would often yield better performance than directly using the raw features if only with proper transformation. We will add these results and discussions in our final version.
>
> Fortunately, there also exists effective solutions for this issue, such as using adaptive partitions/transformations for different features or frequency-based feature grouping [1]. These advanced encoding/embedding techniques can be incorporated into our framework to improve the quality of feature embeddings and we leave it for future work.
>
> **RE**: Permutation invariance aggregation may lose position information
>
> This is an insightful point. In fact, the permutation invariant aggregation can be inherently satisfied by off-the-shelf deep feature learning models, like embedding + vanilla DNN, Wide&Deep, DeepFM, xDeepFM, etc. (See line 122-128 and appendix A for more discussions). Actually, these state-of-the-art methods have not considered the feature position information, and to our knowledge there is littile evidence that such position information would bring up performance gain in learning with attribute features. Maybe more exploration on this point can be done in the future.
>
> The reviewer also raised a point that the discretization would lose information about two 0-1 features from the same raw feature. This is also a good point. To resolve this issue, one can introduce an auxiliary embedding for each raw feature and concatenate or add it with the embedding of each 0-1 feature transformed from one raw feature. Then the auxilliary embedding can learn to capture shared information among the 0-1 features of the same raw feature.
>
> **RE**: (in limitation) The proposed method does not extend to problems in NLP and CV straightforwardly
>
> Thank you for raising this interesting question. As you have noticed, our method can be directly applied to situations with attribute features as input, which are common in recommender systems, advertisement, etc. and classic ML tasks (classification/regression for tabular data). In NLP/CV, there are also scenarios where attribute features are used as (partial) input information, e.g., question answering [2-3]. Our method can also be adopted to these cases for feature extrapolation problem.
>
> Furthermore, we understand that the gap between attribute features and vision/text data lies in the input data format. Yet, since the pretraining + finetuning has become a widely adopted paradigm in CV/NLP areas, one can often have access for pretrained representations before downstream tasks. Therefore, one can treat the representations of images/audio/texts as input for our framework and leverage it for finetuning on specific downstream tasks. In this regard, our method can be extended to handle new information collected from distinct sources/modality or even combine the representations from multiple pretrained networks. We believe these are promising directions and leave them for future exploration.
>
>
> **RE**: Minors
>
> Thanks for pointing out the typos. There is a typo in the caption of Fig. 5 where the unobserved features are marked with yellow color. The standard deviation results (with five trials using different random seeds) for Fig. 4 are presented in Table 5-10 in the appendix and we will incorporate such results in the maintext in the final version. The improvement is statistically significant in most cases.
>
>
>  References:
>
> [1]  Learning Multi-granular Quantized Embeddings for Large-Vocab Categorical Features in Recommender Systems, in WWW2020.
>
> [2] Hybridqa: A dataset of multi-hop question answering over tabular and textual data, In EMNLP2020.
>
> [3] Open question answering over tables and text, in ICLR2021.

---

> > ### Comment · Reviewer_2Jqz · 2021-08-21
> > **Thank you for your response. Some follow-ups questions.**
> >
> > Dear authors,
> >
> > Thank you very much for addressing my concerns! I have no more question regarding the permutation invariance and limitation of applications.
> >
> > And thank you for sharing some additional experimental results regarding the effect of discretization. It is reassuring to see that the accuracy of the three algorithms you experimented with move in the same direction as the bin number changes. (By the way, what's the percentage of observed features used in these additional experiments?)
> >
> > A natural follow-up question, in my opinion, would be: how well does a network which doesn't discretize the continuous feature perform? If we do not care about taking advantage of the new unobserved features, the baseline model could be one trained with features observable during training but without discretization. Is the Base-NN already such a model? If so, please ignore this question.
> >
> > An unrelated question (my apology if this has already been addressed somewhere): do we know how much does the GNN itself contribute to the performance of the proposed model? Suppose we train the GNN + backbone model as proposed in the paper. But, during test time, we still limit ourselves to features observed during training time. How does the accuracy look like? Does the trained GNN play some role of "denoising" the features (or robustify the backbone model) and, hence, lead to a better accuracy on test data? This may also help us understand how much gain is really from the knowledge in the new feature.

---

> > > ### Author Response · Authors · 2021-08-23
> > > **Response to the follow-up questions**
> > >
> > > Thank you for the follow-up comments. In the experiments for bin numbers, we consider observed feature ratio 0.7. Below we provide further experiment results to answer your follow-up questions.
> > >
> > > **RE**: Comparison with directly using continuous features for training.
> > >
> > > The baseline method Base-NN uses transformed features as input (i.e., if there are continuous features in one dataset, we convert them into discrete ones), the same as other methods for fair comparison in our experiments. We ran additional experiments directly using the raw continuous features for Base-NN's input on Protein, Robot, Drive, Calls (The Gene and Github datasets have only discrete features so we omit them). When we also consider observed feature ratio 0.7, we obtained the results for Base-NN using raw features and transformed features:
> > > - Protein: 0.7450 (raw), 0.7462 (transformed)
> > > - Robot: 0.8605 (raw), 0.9071 (transformed)
> > > - Drive: 0.5715 (raw), 0.6603 (transformed)
> > > - Calls: 0.9816 (raw), 0.9840 (transformed)
> > >
> > > As we can see, there is a performance drop when we remove the transformation for continuous features, especially on Robot and Drive. Actually, there are only continuous features in Robot/Drive while in Protein/Calls there exist both continuous and discrete features. The reason is that compared with directly using raw continuous features, the one-hot encodings of discretized features enable the model to consider embeddings for different values that can provide better representation capacity when the transformation is properly considered. The improvement by the representation capacity would become more significance as the instance number goes larger, which is the second reason why the performance variation in Drive is much more. Also, converting continuous features to discrete ones is a common practice that is widely used to handle attribute features in many real-world problems [1,2], especially used as an important component in many data challenges like Kaggle. Therefore, we follow this conventional rule in our setting as we focus on attribute features as input.
> > >
> > > **RE**: Comparison with not using new features during test.
> > >
> > > Thanks for raising this interesting question that can help to improve the quality of our work. We thought about similar questions during our initial exploration, yet we did not run the experiments as is exactly suggested. Here we supplement new experiment results and further discussions to answer your question: how much the gain is from GNN absorbing the new knowledge? We first provide the results on UCI datasets when we do not use new features during test for our model FATE trained with proposed approaches (here we also consider observed feature ratio 0.7):
> > > - Gene: 0.9354 (FATE), 0.9263 (FATE w/o new features), 0.9109 (Base-NN)
> > > - Protein: 0.9490 (FATE), 0.9231 (FATE w/o new features), 0.7462 (Base-NN)
> > > - Robot: 0.9177 (FATE), 0.9105 (FATE w/o new features), 0.9071 (Base-NN)
> > > - Drive: 0.6704 (FATE), 0.6511 (FATE w/o new features), 0.6603 (Base-NN)
> > > - Calls: 0.9946 (FATE), 0.9921 (FATE w/o new features), 0.9840 (Base-NN)
> > > - Github: 0.8749 (FATE), 0.8701 (FATE w/o new features), 0.8664 (Base-NN)
> > >
> > > The results show that the GNN module and our proposed training method helps for learning with observed features ("FATE w/o new features" outperforms Base-NN in most cases). Yet in some cases (e.g., Drive), "FATE w/o new features" gives inferior performance than Base-NN. The results indicate that our additional GNN module and self-supervised training approach brings up some regularization effects. Further interpretation for such results can be reflected by Fig. 5 in our maintext where the visualized embeddings show that our introduced GNN module can leverage locality structures to encode more feature-level relations, contributing to learning better feature representation than directly training the backbone (i.e., Base-NN). Moreover, notably, our model FATE consistently outperforms "FATE w/o new features" through these datasets, which proves that the proposed approach can indeed help for absorbing the knowledge from new features.
> > >
> > > Secondly, it is worth noting that our feature extrapolation problem generally entails two settings: 1) new raw features appear; 2) new values out of the known space of existing raw features appear (as illustrated in line 100-101). Our experiments on Avazu and Criteo are designed for the second case and we split the dataset in chronological order to simulate real-world scenario, which guarantees that in the test splits we have new values (i.e., 0-1 features) out of the known space of raw features in the training split (the ratio of new/old 0-1 features is about 1:1). We observe that the performance would degrade obviously if we do not use the embeddings for new features given by our GNN module (i.e., set the embeddings for unseen features as zero for inference during test). Concretely, the overall AUCs with DNN as backbone degrade from 0.710 to 0.690 on Avazu and from 0.772 to 0.766 on Criteo (a improvement of 0.005 for AUC is considered significant in CTR prediction task). Therefore, we can see that our model produces superior power for feature value extrapolation, which can be another important application aspect of our approach.
> > >
> > > We hope the new results and discussions help to further strengthen our contribution. Thank you again for your thorough review and please let us know if you have any further comment or question.
> > >
> > > Reference
> > >
> > > [1] Field-aware Factorization Machines for {CTR} Prediction, in RecSys 2016.
> > >
> > > [2] Filed-wise Learning for Multi-filed Categorical Data, in NeurIPS 2020.

---

> > > > ### Comment · Reviewer_2Jqz · 2021-08-25
> > > > **Thank you for sharing the new results**
> > > >
> > > > Thank you for taking the effort to perform these extra evaluations. Theses new results addressed my questions very well.

---

### Official Review · Reviewer_fz9M · 2021-07-17

**Rating:** 4
**Confidence:** 4

**Summary:**

This paper studies the “open-world” feature extrapolation problem and develops a neural model that can extrapolate to new features without any retraining. They provide a graph model for this problem, and a few standard theoretical results.

**Limitations And Societal Impact:**

Yes

**Main Review:**

The idea in this work is interesting, though the actual setup comes with many assumptions and limitations. It seems the contribution of this work is more on the problem side, as the proposed approach is simply based on standard methods. Also, there are some works that seem closely related in spirit, “Latent Graph Predictor Factorization Machine (LGPFM) for modeling feature interactions weight”, “Deep Relational Factorization Machines”, “Feature Interaction-aware Graph Neural Networks”. There are likely others that are also missing from the related work, and should be discussed, and the differences clarified. What is the time and space complexity? And how does it compare to other related methods?

**Time Spent Reviewing:**

3

---

> ### Author Response · Authors · 2021-08-08
> **Response to Reviewer fz9M**
>
> We thank the reviewer for acknowledging our contribution on problem setting side. The major concerns lie in our contribution on methodology side. However, we believe that there is a misunderstanding regarding our method (as suggested by the references [1-3] given by the reviewer) and the reviewer may underestimate the technical challenges of our setting. In the response below, we first clarify our differences compared with related works, highlight the challenges under our setting and further discuss its impacts. After that, we address the concerns about limitations/assumptions and complexity. We hope our response help to better elaborate our contributions and significance.
>
> **RE**: Originality, significance and differences to suggested references
>
> The reviewer pointed out that our method is closely linked to [1-3] which essentially consider feature interactions through learning and message passing on graph among attribute features via self-attentive graph predictor [1], a predefined concurrence matrix [2] or instance-specific attention [3]. In fact, while these works seem related, the key points of their models/training are different to ours and they are designed for standard problem settings which are much simpler than us and have constrained scopes. We next provide a detailed illustration.
>
> - **Distinct model.** In [1-3] and other works focusing on feature interactions, the model takes an instance $x=(x_1, x_2, ..., x_d)$ (with fixed length) as input and considers interactions among any feature pair $(x_i, x_j)$ to obtain $x$'s representation. Since the feature space is shared by training and test data, the feature embeddings can be reused and the GNN model is adopted to **aggregate features' embeddings** of the neighbors in **a static feature graph** in order for better representation capacity. By contrast, in our setting, the input instance's features will be augmented in the future and new features in the test data would have no available embeddings. The GNN module aims to leverage embeddings of existing features to estimate embeddings of new ones in an inductive manner. The GNN serves as a role for **transferring knowledge** in existing concepts to new unseen ones that come from **dynamic open world**, **instead of merely fitting a mapping from x to y in a static dataset** like [1-3].
>
> - **Distinct training approach.** In [1-3], the model is trained in a standard supervised setting where the distributions for training and test data are assumed to be the same and the trained model cannot handle new unseen features in the future. Differently, our model is trained via our proposed training approaches (one by inductive learning and one by self-supervised learning), where we split/sample partial observed features as proxy data and update the backbone and GNN module with asynchronous speeds. These new techniques are crucial parts for enabling feature extrapolation, which is also our contribution.
>
> - **Unique challenges.** The core challenge of our problem open-world feature extrapolation is how to bridge the relationship with seen and unseen features for achieving **out-of-distribution generalization without re-training on new data**. Previous methods would fail on this issue since they assume the feature space is shared by training and test data. Therefore, our model is not a trivial extension of GNN-based supervised learning. Moreover, one important research question of our problem is the generalization ability of proposed model, for which we do theoretical analysis in Section 3 based on our setting and training approach. Also, such analysis is not a trivial extension from existing studies.
>
> - **Impacts.** Our work serves as the first attempt for open-world feature extrapolation which is an important problem for modern ML systems that need to handle new observation unseen before. Furthermore, our GNN-based knowledge transfer explores a new perspective for representation learning via inductive reasoning from existing concepts (abstracted by the backbone network) to new ones in-the-wild.
>
> **RE**: "The setup comes with many assumptions and limitations"
>
> We sincerely hope the following discussion with supporting evidence could help for your re-assessment of our work. The problem we formulate in this paper assumes attribute features as input and the feature space would go through expansion in the future. Therefore, our method can be applied to **arbitrary scenarios with attribute features as input**, e.g. learning from tabular data, recommender systems, online advertisement display, question answering, etc. to tackle the dynamic feature space expansion which is common in real-world scenarios where new information is incrementally collected. As for our method, there are two modules in our proposed framework: one is a backbone network and one is a GNN network. The GNN module acts as plug-in components and **one can instantiate the backbone as off-the-shelf neural architectures for deep feature learning**, like DNN, Wide&Deep, DeepFM, xDeepFM, etc. For example, in our experiments on Criteo and Avazu, we adopt DNN and DeepFM as the backbone and achieve promising results. Besides, as discussed in Section 5, the framework can also be extended to multi-modal learning and federated learning where features are incrementally collected from distinct fileds. **The assumptions w.r.t. model architecture and training algorithm are only made for theoretical analysis**, and they are not strong ones (See our Section 3). **When it comes to practice, these assumptions are not necessary**. Our experiments on eight datasets from distinct domains (biology/engineering/social/advertisement) with varied sizes (from ~1K to ~40M instances) and diverse feature types (discrete/continuous/mixture) also prove the wide applicality of our model.
>
> **RE**: What is the time/space complexity
>
> We agree that a scalable model would be more practically useful for large-scale data. Our GNN model is operated on a feature-instance bipartite graph of $B$ instance nodes and $|F_s|$ feature nodes (where $B$ is the size of mini-batch instances and $|F_s|$ is the size of sampled features). The edge number in the graph would be no more than $B\times d$ (see discussion in line 189-197). Thus, the time/space complexity is $O(B\times d\times L\times H^2)$ where $L$ is the GNN's layer number and $H$ is the hidden size. In practice, we use $L\leq 4$, so the time/space cost mainly depends on the sizes of training instances and features that **contribute to linear scalability, which is validated by our empirical time/space scalability test in Fig. 6 and 7** in appendix. Our experiments on two large datasets Criteo and Avazu (with ~40M instances and ~2M features and we set batch size as 1e5) require less than 12GB GPU memory for training/inference, which also prove that our model can scale smoothly.
>
> We next compare with [1-3] suggested by the reviewer. [1] needs to deal with fully-adjacent graphs among features which induces $O(B\times d^2\times L \times H^2)$. [2] requires time/space complexity $O(B\times d^2\times H^2 + |\mathcal E|\times L\times H^2)$ where $|\mathcal E|$ is the edge number in the predefined feature co-occurrence graph and $d\leq|\mathcal E|\leq d^2$. Also, [3] requires $O(|\mathcal E|\times L\times H^2 + B\times d^2\times H^2)$. Therefore, **the complexity of our approach is at most comparable to these methods**. Again they are limited in standard settings while our model is designed for OFE problem.
>
> We hope the response helps for addressing your concerns and please let us know if you have any further comment or question.
>
> References:
>
> [1] Latent Graph Predictor Factorization Machine (LGPFM) for modeling feature interactions weight, in SITA 2020
>
> [2] Deep Relational Factorization Machines, in openreview 2019
>
> [3] Feature Interaction-aware Graph Neural Networks, in Arxiv 2020

---

> ### Author Response · Authors · 2021-08-23
> **Thank you for the time and hope our response and positive feedbacks from other reviews help for your re-assessment of our work.**
>
> Dear Reviewer fz9M,
>
> We notice that in the initial review, your concerns lie in three parts. 1) The main concern is our contribution on methodology side, for which we provide a clear comparison to the suggested related works (see response below). 2) The secondary concern is the limitation of our setup, yet similar points have been positively recognized by the feedbacks from Reviewer T7WN and 2Jqz. 3) The third question is about time/space complexity, for which we believe the posted response below with supporting evidence in our paper can address your concern.
>
> Given these facts and positive feedbacks from other three reviewers, we sincerely hope you could re-consider your initial rating. Also if you have any futher comment or question, please let us know and we are glad to provide follow-up response.

---

> ### Author Response · Authors · 2021-08-31
> **Look forward to feedbacks**
>
> Dear Reviewer fz9M,
>
> Thanks for your initial comments. We sincerely hope you can give us some feedbacks and further comment based on our provided response and review comments from other reviewers. We would like to know whether our response helps for addressing your misunderstanding and the re-assessment of our work.

---

> ### Author Response · Authors · 2021-09-01
> **Thank you for the time and we would like to see if there is any further concern**
>
> Dear Reviewer fz9M,
>
> Since it is approaching the end of the discussion period, we would like to kindly ask if our previous response clarifies your concerns and if there are any further questions that we could answer to facilitate the review process. Thanks a lot for your time!

---

### Official Review · Reviewer_tu7i · 2021-07-18

**Rating:** 6
**Confidence:** 3

**Summary:**

This paper introduces the “open-world feature extrapolation” setting where models are trained on a set of features but introduced to additional features as time goes on or at deployment. This is a reasonable assumption in practice, as ML frameworks in practice are constantly updated with new items (such as recommender systems with new products, new users). Naively, one could retrain on the existing features and the new features, but this is time-consuming and infeasible for online settings; one could also retrain on solely the new features but this poses the risk of forgetting existing features or overfitting.

To address this, the paper introduces FATE, a framework that can handle new features at test time without retraining the model. A neural network can be viewed as an embedding layer plus a classifier. The embedding layer maps the input features x to a H-dimensional hidden vector z, so the technical challenge is to find a weight embedding (for x to z) for new features, given existing features. The paper uses GNN message-passing over a feature-data graph (feature nodes and instance nodes) to learn these. To train the model, two approaches are presented – self-supervised learning where a portion of the features are withheld and whose embeddings are estimated by the model using the other features; and inductive learning where a k-subset of the features are sampled.

The paper presents a theoretical result on the generalization error (under the inductive learning approach) as well as experiments comparing against baselines that either incorporate the new features in other ways or do not at all.

**Ethical Concerns:**

I do not see any ethical concerns with this submission.

**Limitations And Societal Impact:**

The authors have adequately addressed limitations and societal impacts.

**Main Review:**

Originality: this paper’s setting is distinct from other settings with shifts and expansions in distributions and/or their support. It does seem similar to the cold start problem in recommendation systems and “tail” entities in problems like named entity disambiguation  (perhaps with a stricter requirement of no retraining, vs in general how to handle rare items). In all of these cases there are some “features” that are rare and thus may not be seen at training. In the NLP space, something like [1]  aims to infer embeddings of rare words given embeddings of frequent words, while [2] tackles the tail by leveraging additional reasoning patterns. There has also been work in the zero-shot setting [3, 4], and in recommendation systems [5] uses a bipartite user-item graph and GNNs for the cold start problem. I thus recommend having more related work in the above vein comparing both methods and motivation.

[1] Timo Schick and Hinrich Schütze. 2020. Rare words: A major problem for contextualized embeddings and how to fix it by attentive mimicking. In Proceedingsof the AAAI Conference on Artificial Intelligence, Vol. 34. 8766–8774.

[2] Laurel Orr, Megan Leszczynski, Simran Arora, Sen Wu, Neel Guha, Xiao Ling,and Christopher Re. 2020. Bootleg: Chasing the Tail with Self-Supervised Named Entity Disambiguation. arXiv preprint arXiv:2010.10363(2020).

[3] Lajanugen Logeswaran, Ming-Wei Chang, Kenton Lee, Kristina Toutanova, Ja-cob Devlin, and Honglak Lee. 2019. Zero-shot entity linking by reading entity descriptions. arXiv preprint arXiv:1906.07348(2019).

[4] Tao Wu, Ellie Ka-In Chio, Heng-Tze Cheng, Yu Du, Steffen Rendle, Dima Kuzmin,Ritesh Agarwal, Li Zhang, John Anderson, Sarvjeet Singh, et al. 2020. Zero-Shot Heterogeneous Transfer Learning from Recommender Systems to Cold-Start Search Retrieval. In Proceedings of the 29th ACM International Conference on Information & Knowledge Management. 2821–2828.

[5] Bowen Hao, Jing Zhang, Hongzhi Yin, Cuiping Li, and Hong Chen. 2021. Pre-training Graph Neural Networks for Cold-Start Users and Items Representation. In Proceedings of the 14th ACM International Conference on Web Search and Data Mining. 265-273.

Quality: I looked over the theory and experiments quickly and most of it appeared correct. I have a few questions about the generalization error result.
1) Theorem 1 describes the generalization error on the test data with new features. I am guessing this is reflected through the expectation in eq 4 being over the feature set F, in addition to (X, Y). However, I don’t completely understand how the proofs in the Appendix address randomness over F.
2) Why do the bounds not scale with the sample size for each X_m? (i.e. where is |I| or N?)
3) Why does the stability parameter \gamma increase with more iterations? It seems unusual to me that the dependency on d scales exponentially with the number of iterations T.
4) Theorem 1 and Table 4 suggest that the number of raw features k plays an important role, but it is not clear where k is in the bound.


Clarity: this paper was enjoyable to read aside from a few grammatical errors that can be fixed easily.

Significance: the paper tackles a significant problem. As we think about models “in-the-wild” that deal with dynamic data, it is crucial to understand how to continuously incorporate new information efficiently and accurately.


**Time Spent Reviewing:**

10

---

> ### Author Response · Authors · 2021-08-08
> **Response to Reviewer tu7i**
>
> We thank the reviewer for appreciating our motivation/presentation and acknowledging our distinct problem setting that potentially have significant impact especially for in-the-wild generalization. The reviewer asked us to further discuss relationship with several suggested related works and raised several questions regarding the analysis. Below we address the concerns in order to better elaborate our contribution.
>
> **Comparison with zero-shot/few-shot learning and cold-start recommendation** Our work is closely linked to the works suggested [1-5] but has its own uniqueness.
>
> On **problem setting** side, we briefly discuss the differences from three aspects. First, we focus on expansion of input space, different from zero-shot/few-shot learning, like entity disambiguation [2] and entity link prediction [3] that deals with rare/unseen targets from output space . Second, we disallow any re-training on new data and the new features from expanded space are strictly unseen without any side information, in contrast to few-shot learning cases [1, 2] where the 'tail' words/entities are seen in training set and the models using side information, e.g., entity dictionary [3] and item-item graphs [4], that helps zero-shot inference. Third, in our setting, the distribution shift of input space lies in expansion of the feature space, which poses challenges of how to enable a model trained on instances with partial features (as input attributes) to generalize to test instances with more features, different from cold-start recsys [5] where the new objects, i.e. users or items, correspond to 'instances' instead of features. There are also feature-based recommendation models, yet they mostly focus on static feature space even when considering cold-start users/items. Indeed, our problem open-world feature extrapolation (OFE) is orthogonal to the above works and opens a new direction. Also, it could be promising to combine OFE with the above settings, which can be potential future directions.
>
> On **methodology** side, the key originality of our model is to use a set of features' embeddings as seeds to represent others via message passing over a feature-instance graph induced by the data matrix. Essentially, such a design can be understood as an explicit approach for out-of-distribution generalization, mimicking a reasoning process from familiar concepts to new ones. To our knowledge, this is also a brand new method in this sense.
> We will incorporate these discussions and related works in our final version.
>
> **Clarification for generalization error analysis.** We next provide a detailed response to the questions in order.
>
> **RE**: How do we address the randomness of observed features in analysis
>
> As the reviewer pointed out, our generalization analysis for test data with new features is based on the expectation over $(\mathbf X, Y)$ in Eqn. 4. It is worth noting that the sampling process for $(\mathbf X, Y)$ includes two steps: 1) sampling a feature set $F= ${ $f_j$ } according to $f_j\sim \mathcal F$ and 2) sampling data matrix $(\mathbf X, Y)=(${$\mathbf x_i$}, {$y_i$}$)$ according to $(\mathbf x_i, y_i)\sim \mathcal D_{F}$. Therefore, the expectation is taken over two distributions, which covers randomness of both feature sampling and data generation. Throughout our proof in the Appendix, we simplify the notation as $\mathbf O = (\mathbf X, Y)$ and the generation process for $\mathbf O$ also includes first sampling $F$ and then sampling $(\mathbf X, Y)$ conditioned on $F$. Therefore, the expectation over $\mathbf O$ (e.g., in Eqn. 17, 18 and 19) is taken over $p(F)$ and $p(\mathbf X, Y|F)$, considering the randomness of both feature sampling and data generation.
>
> **RE**: Why do the bound not scale with size of training data $N$
>
> This is a good question. We agree that generalization error bound often depends on input sample size. In our case, this holds true though in slightly different forms. First, to stay consistent with the training approach which samples data sub-matrices from original training data for each update, we treat the data sub-matrix $(\mathbf X_m, Y_m)$ as 'data points' for analysis. In this case, the corresponding sample size would be $M$ (by our definition). As shown in Eqn. 7, when $M$ becomes larger, the bound will be tighten, which accords with the intuition. Furthermore, the bound also scales with $N$ if using mini-batch instance-level partition since $M$ depends on $N$. Specifically, we have $M = { N\choose B} \times { d\choose k}$ (see line 570 of the appendix).
>
> **RE**: Why does the stability parameter $\gamma$ increase exponentially with more iterations
>
> In fact, this is due to the expectation of error accumulation (which is determined by $d$ as in Eqn. 27 and 28) through multiple SGD iterations, as quantitatively shown in Eqn. 29. Intuitively speaking, with more iterations, the variance of sampling would be enlarged and each iteration will contribute to the probabilities that two algorithms sample different data points in any of the position during the training dynamics. Similar trends can be found in the Thm 3.12 of [18] and Thm 3 of [44], which also justify the result in our paper.
>
> **RE**: What is the role of sampling size $k$ in the bound
>
> The number of sampled features, i.e. $k$, indeed plays a role in the bound since it impacts the randomness of sampling data-matrices and the value $M$ (see line 218). Concretely, we have $M = { N\choose B} \times { d\choose k}$ when using mini-batch instance-level partition and $M = { d\choose k}$ otherwise (see line 562-571 in appendix).
>
> Please let us know if you have any further questions and comments.
>
>
>
> References:
>
> [18] Train faster, generalize better: Stability of stochastic gradient descent, in ICML 2016.
>
> [44] Stability and generalization of graph convolutional neural networks, in KDD 2019.

---

> ### Author Response · Authors · 2021-08-25
> **More comparison on method and motivation**
>
> Thank you for the comments. We supplement more discussion comparing our work with few-shot/zero-shot learning in NLP [1-3] and cold-start recommendation [4,5] on the method and motivation. As you mentioned, in some NLP tasks, there are rare entities exposed in limited times or new entities unseen by training; in cold-start recsys, there are also new users/items unseen before. In these tasks, one challenge is how to obtain the representation for new objects. Our method shares some high-level conceptual thinking but has several notable differences, especially under our unique problem setting. Below we provide detailed illustration.
>
> **How to model the relations between existing/new objects**. We focus on learning mapping from input attribute features of an instance to the label, based on which we can treat observed data, i.e. an instance-feature matrix as a bipartite graph. The key insight is that the feature-instance graph (which can be augmented in the future due to new features and instances) explicitly embodies the promixity and locality structures among features (with instances treated as the intermediate nodes), on top of which we can leverage a GNN module for transfering knowledge from existing features to new ones via message passing. While [1-5] also aims at inferring the embeddings for new entities/users/items based on some 'held-out' ones, our ways for achieving the goal is different compared to them, e.g. [1] using concurrence of rare/frequet words, [2] using additional reasoning pattern, [3] using entity description, [4] using predefined item-item correlation graph or [5] using observed user-item graph, to bridge the exisitng and new objects of interests.
>
> **How to train the model for extrapolation**. We propose two training approaches to endow the model with ability for feature extrapolation, one by self-supervised learning and one by inductive learning. The key insight backed up with our theoretical analysis is to simulate new features during training by partioning or sampling over observed features to construct proxy training data matrix, which are specified in different ways in the two approaches. This strategy shares some high-level spirits with the designs in [2] and [5] which respectively simulates new domains and cold-start users/items by domain-adaptive pre-training and reconstruction. Yet our ways for achieving the goal are quite different under our unique problem setting with distinct technical challenges.
>
> We appreciate your time and hope our two sets of response help to properly address your questions

---

> ### Author Response · Authors · 2021-08-31
> **Look forward to feedbacks**
>
> Dear Reviewer tu7i,
>
> Thanks for your thorough initial comments. We would like to know whether our response has addressed your questions properly. As the discussion period will end soon, we sincerely hope to receive your further feedbacks, and we can provide follow-up response if needed.

---

> > ### Comment · Reviewer_tu7i · 2021-08-31
> > **Thank you for your response**
> >
> > Thank you for your response regarding my questions about the related work and theory. I would like to see this paper accepted if the related work is updated to thoroughly compare against these similar problem settings mentioned.

---

> > > ### Author Response · Authors · 2021-08-31
> > > **Thanks for your feedbacks and constructive suggestions**
> > >
> > > Thank you for the feedbacks and constructive suggestions. We will incorporate the discussion on these related works in our final version where the one additional page allows us to extend the current Section 5 with more content and illustration. Also, for your quick check, we write a draft version below that we plan to insert into Section 5 of our main paper.
> > >
> > > *Our problem setting is also linked with few-shot/zero-shot learning. In NLP domains, some studies focus on dealing with rare entities exposed in limited times or new entities unseen by training [1,2,3]. Similar problems are also encoutered and explored in cold-start recommender systems where there are also new users/items unseen before [4,5]. One common nature of these works aim at inferring the embeddings for new entities/users/items based on some 'held-out' ones, via e.g. [1] using concurrence of rare/frequet words, [2] using additional reasoning pattern, [3] using entity description, [4] using predefined item-item correlation graph or [5] using observed user-item graph, to bridge the exisitng and new objects of interests.*
> > >
> > > *By contrast, our work is distinct from them in several aspects. On problem setting side, our OFE problem focus on expansion and distribution shift of input feature space. This is different from entity disambiguation [2] or entity link prediction [3] that handle rare/unseen targets in output space, and different from new users/items that correspond to new `instances' rather than features. There are also feature-based recommendation models, yet most of them assume static feature space. Therefore, our setting is orthogonal to these cases. Second, on methodology side, we resort to a feature-instance graph that bridges the new and existing features and GNN's message passing for feature embedding extrapolation, which deviates from the main body of these works. Also, our proposed two training approaches share some high-level spirits with self-supervised learning in [2] and [5] that simulates new concepts for training, yet our ways for achieving the goal (via feature sampling/partition for proxy data matrix) are quite different under our unique problem setting with distinct technical challenges.*
> > >
> > > We appreciate your time and please let us know if you have further comment.

---

### Decision · Program_Chairs · 2021-09-27

**Decision:**

Accept (Poster)

**Comment:**

Overall, this paper’s reviews have reached a consensus. Three of the reviewers wish for the paper to be accepted, in particular after some high-quality clarifications from the authors. I agree with this decision. There is only one reviewer who is opposed, but that reviewer’s disagreement is based on a potential similarity between the paper and previous work, which the authors have successfully dispelled. I assume the reviewer is comfortable with this response; in any case, there are clearly sufficient distinctions from prior work and sufficient novelty here.

From my own read, I think this paper identifies a fairly neat problem setting (ie, feature space expansion at test time) and makes it concrete, then offers reasonable solutions to it. The problem is a practical one, but it hasn’t been tackled in this exact form before (at least, to the best of my knowledge). While theoretically the problem could be posed in a far more general framework on domain shift, it would be hard to find a good solution. The current identification leads to some pretty nice results, and it could serve to drive a bunch of new research questions and solutions.